# SAFE CONTINUOUS-TIME MULTI-AGENT REINFORCEMENT LEARNING VIA EPIGRAPH FORM

**Xuefeng Wang**[1][*] **Lei Zhang**[1][*] **Henglin Pu**[1], **Husheng Li**[1][†] **Ahmed H. Qureshi**[1][†]
[1]Purdue University

## ABSTRACT

Multi-agent reinforcement learning (MARL) has made significant progress in recent years, but most algorithms still rely on a discrete-time Markov Decision Process (MDP) with fixed decision intervals. This formulation is often ill-suited for complex multi-agent dynamics, particularly in high-frequency or irregular time-interval settings, leading to degraded performance and motivating the development of continuous-time MARL (CT-MARL). Existing CT-MARL methods are mainly built on Hamilton–Jacobi–Bellman (HJB) equations. However, they rarely account for safety constraints such as collision penalties, since these introduce discontinuities that make HJB-based learning difficult. To address this challenge, we propose a continuous-time constrained MDP (CT-CMDP) formulation and a novel MARL framework that transforms discrete MDPs into CT-CMDPs via an epigraph-based reformulation. We then solve this by proposing a novel physics-informed neural network (PINN)-based actor–critic method that enables stable and efficient optimization in continuous time. We evaluate our approach on continuous-time safe multi-particle environments (MPE) and safe multi-agent MuJoCo benchmarks. Results demonstrate smoother value approximations, more stable training, and improved performance over safe MARL baselines, validating the effectiveness and robustness of our method. Code is available at this link.

## 1 INTRODUCTION

MARL has achieved remarkable success in diverse domains, ranging from strategic games (Samvelyan et al., 2019; Vinyals et al., 2019), multi-robot coordination (Haydari & Yılmaz, 2020; Kuyer et al., 2008), and wireless communication (Wang et al., 2023). These advances demonstrate the potential of MARL as a powerful framework for solving complex cooperative and competitive decision-making problems. Despite these achievements, most existing MARL algorithms are formulated in discrete time and fundamentally rely on the Bellman equation (Bellman, 1966). This formulation often assumes fixed time intervals between decision steps, which is adequate in settings where the decisions naturally occur at uniform time intervals. However, this assumption is not well-suited for complex high-frequency domains such as autonomous driving (Kiran et al., 2021; Chen et al., 2021), financial trading (Shavandi & Khedmati, 2022), where decision-making requires continuous-time control. In such cases, discrete-time RL often struggles to learn accurate policy (Doya, 2000; Mukherjee & Liu), as fixed-step discretization fails to represent non-uniform temporal dynamics, resulting in degraded performance and unstable learning (Tallec et al., 2019; Park et al., 2021; De Asis & Sutton, 2024). These limitations highlight the necessity of developing an alternative framework beyond discrete-time Bellman equations, which is compatible with CT-MARL.

Recent studies (Wang et al., 2025) have explored the HJB equations to solve CT-MARL problems. The HJB can be viewed as the continuous-time analogue of the Bellman recursion, where the value function is characterized as the viscosity solution of a nonlinear Partial Differential Equation (PDE) (Shilova et al., 2024). In practice, PINNs have emerged as a common approach to approximate HJB solutions: they train neural networks to minimize HJB PDE residuals and leverage gradient-consistent signals for policy improvement (Mukherjee & Liu; Meng et al., 2024). This formulation eliminates the need for fixed time discretization and enables MARL to operate in continuous-time

---

[*]Equal contribution.
[†]Corresponding author.

domains. However, in safety CT-MARL settings, state constraints (e.g., when they are treated as collision penalties) introduce value discontinuities, making it difficult for HJB-based PINNs to approximate the value functions accurately (Zhang et al., 2024).

To address these challenges, we first cast safe CT-MARL as a CT-CMDP with explicit state constraints. We then introduce a revised epigraph reformulation that augments the system with an auxiliary state $z$, transforming the discontinuous constrained values into a continuous form suitable for PDE-based learning. On top of this reformulation, we adopt an actor–critic framework to learn values and policies under continuous-time state constraints. Specifically, we improve epigraph-based training by integrating the inner and outer optimization into a unified scheme. At each rollout, we compute the optimal auxiliary state $z^*$ and uses it directly for training, while keeping all networks $z$-independent. This design avoids the noise of random $z$ sampling, yields more accurate policy updates, and eliminates costly root-finding at execution.

Our main contributions are summarized as follows. **(1)** To the best of our knowledge, this is the first work to explicitly incorporate state constraints into the formulation of CT-MARL. We introduce an epigraph-based reformulation to bounds discounted cumulative cost and state constraints within a unified objective, effectively transforming discontinuous values into continuous ones. **(2)** We design an improved epigraph training scheme that integrates inner and outer optimization, providing more stable learning signals and removing the need for costly root-finding algorithms. **(3)** We prove the existence and uniqueness of viscosity solutions for epigraph-based HJB PDEs, providing theoretical support for our method. Extensive experiments on adapted continuous-time safe MPE and multi-agent MuJoCo benchmarks further demonstrate that our approach consistently outperforms current safe MARL methods.

## 2 RELATED WORK

### 2.1 CONTINUOUS-TIME REINFORCEMENT LEARNING

Discrete-time reinforcement learning (DTRL) often performs poorly in continuous-time environments, particularly when decision intervals are irregular (Tallec et al., 2019; Park et al., 2021; De Asis & Sutton, 2024). Consequently, continuous-time reinforcement learning (CTRL) has received growing attention as a more suitable framework for such problems (Doya, 2000; Yildiz et al., 2021; Wang et al., 2020; Bradtke & Duff, 1994; Jia & Zhou, 2022a;b). Most existing studies focus on the single-agent setting, proposing various approaches for value function approximation (Mukherjee & Liu; Wallace & Si, 2023; Lee & Sutton, 2021). For example, Mukherjee & Liu employ PINNs to approximate the value function and guide a PPO-based policy update, while Jia & Zhou (2022b) address stochastic dynamics through a Martingale loss designed for stochastic differential equations. In contrast, research on CT-MARL remains limited. Prior works (Luviano & Yu, 2017; Jiang et al., 2023) have considered multi-agent problems in continuous time, but largely in application-specific contexts rather than as general-purpose algorithms. The study in Wang et al. (2025) represents the first systematic attempt to design CT-MARL methods, combining PINNs with value gradient iteration to improve value approximation and performance. However, these approaches still inherit the limitations of PINNs that they can only approximate smooth value functions and therefore neglect safety constraints.

### 2.2 MULTI-AGENT SYSTEMS WITH SAFETY CONCERNS

Multi-agent scenarios often raise critical safety concerns, and directly learning under combined reward and safety signals poses significant challenges. A number of studies have explored safe MARL frameworks to address these issues (Gu et al., 2023; ElSayed-Aly et al., 2021; Gu et al., 2024; Shalev-Shwartz et al., 2016). For instance, Chow et al. (2018) employ primal–dual methods to enforce safety constraints, while Althoff et al. (2019) adopt a trust-region approach. Gu et al. (2021) introduce MACPO and MAPPO-Lagrange, which provide theoretical guarantees for both monotonic reward improvement and safety constraint satisfaction. In addition, Zhang et al. (2025b) leverage epigraph forms to formulate multi-agent safe optimal control problems, improving stability during training. However, these approaches are primarily developed in discrete-time settings, which limits their ability to capture continuous-time dynamics. Some efforts have incorporated safety into continuous-time multi-agent systems (e.g., Tayal et al. (2025)), but they assume fully known system

dynamics and rely on optimal control algorithms, significantly restricting applicability. In more realistic scenarios, where dynamics are only partially known or highly complex, such methods fail to provide practical solutions.

Existing methods remain limited in handling discontinuities and safety constraints in CT-MARL. Discrete-time safe MARL algorithms provide theoretical guarantees but do not naturally extend to continuous dynamics, while continuous-time approaches struggle with discontinuous value functions. To address these challenges, we propose an epigraph-based reformulation that unifies safety constraints and standard cost functions within a single objective, enabling principled and stable learning in CT-MARL.

## 3 METHODOLOGY

In this section, we present our epigraph-based PINN actor–critic iteration (EPI) for solving CT-MARL with state constraints. **1) We first formalize the learning problem as CT-CMDP**. Secondly, **2) we reformulate the CT-CMDP using an epigraph form**. By introducing an auxiliary state $z$ to augment system states, this reformulation converts discontinuous value functions into continuous ones. Building on this reformulation, **3) we develop an actor-critic learning architecture that aligns with the epigraph inner-outer optimization scheme**. Specifically, the outer optimization computes the optimal auxiliary state $z^*$ along the rollout, ensuring that the critic captures the tightest feasible trade-off between return and safety. Based on this, the inner optimization trains the critic using PINNs, which jointly update the return and constraint networks together with $z^*$ to approximate the epigraph-based value function. This stabilized critic then serves as the foundation for actor training: we derive an advantage function consistent with the epigraph-based HJB PDEs, which provides the key learning signal for policy improvement.

### 3.1 PROBLEM FORMULATION

#### 3.1.1 CONTINUOUS-TIME CONSTRAINED MARKOV DECISION PROCESS

We consider a CT-CMDP problem, formally defined by the tuple

$$\mathcal{M} = \langle \mathcal{X}, \{\mathcal{U}_i\}_{i=1}^N, N, f, \{l_i\}_{i=1}^N, c, \{t_k\}_{k\geq 0}, \gamma \rangle, \tag{1}$$

where $\mathcal{X} \subseteq \mathbb{R}^n$ is the global state space, and $\mathcal{U} = \mathcal{U}_1 \times \cdots \times \mathcal{U}_N \subseteq \mathbb{R}^m$ is the joint control space for $N$ agents. The system evolves according to time-invariant nonlinear dynamics $\dot{x}(t) = f(x(t), u(t))$ with $x(0) = x_0$, where $f : \mathcal{X} \times \mathcal{U} \to \mathcal{X}$. Each agent $i$ applies a decentralized policy $\pi_i : \mathcal{X} \times [0, \infty) \to \mathcal{U}_i$, and the joint policy is denoted as $\pi = (\pi_1, \ldots, \pi_N)$. All agents share the non-negative cost function $l = \sum_{i=1}^N l_i$, where $l_i : \mathcal{X} \times \mathcal{U}_i \to \mathbb{R}$ is the independent cost function of agent $i$. The system is further subject to state-dependent safety constraints specified by a function $c : \mathcal{X} \to \mathbb{R}$, with the feasible set defined as $\mathcal{F} = \{x \in \mathcal{X} \mid c(x) \leq 0\}$. Control actions are updated at irregular decision times $\{t_k\}_{k\geq 0}$, with strictly positive intervals $\tau_k = t_{k+1} - t_k$. $\gamma \in (0, 1]$ is the discount factor. Throughout the paper, we assume that $\mathcal{U}_i$ is compact and convex, $f$ and $c$ are Lipschitz continuous, and $l_i$ is Lipschitz continuous and bounded. The joint objective is to minimize the cumulative cost under joint control input $u = (u_1, \ldots, u_N)$ subject to state constraints $c(x)$

$$v(x) = \min_{u \in \mathcal{U}} \int_t^\infty \gamma^{\tau-t} l(x(\tau), u(\tau)) \, d\tau \tag{2}$$
$$\text{s.t.} \quad c(x(\tau)) \leq 0, \quad \forall \tau \geq t.$$

#### 3.1.2 EPIGRAPH REFORMULATION

The value becomes discontinuous (Altarovici et al., 2013) when state constraints are violated in Eq. 2, which hinders the convergence of HJB-based PINN training. To address this, we leverage an epigraph reformulation that converts value in Eq. 2 into a continuous representation.

**Definition 1** (Epigraph Reformulation). We introduce an auxiliary state variable $z(t) \in \mathbb{R}$ to reformulate Eq. 2 using the epigraph forms. Here, $z$ follows the dynamic $\dot{z}(t) = -l(x(t), u(t)) - \ln \gamma \cdot z(t)$. Therefore, the auxiliary value function is defined as

$$V(x, z) = \min_{u \in \mathcal{U}} \max \left\{ \max_{\tau \in [t, \infty]} c(x(\tau)), \int_t^\infty \gamma^{\tau-t} l(x(\tau), u(\tau)) d\tau - z \right\}, \tag{3}$$

**Lemma 3.1** (Value Equivalence). Suppose the assumptions in Sec. 3.1.1 hold. For all $(t, x, z) \in [0, \infty) \times \mathcal{X} \times \mathbb{R}$, the constrained value $v$ and auxiliary value $V$ are related by

$$v(x) = \min\{z \in \mathbb{R} \mid V(x, z) \le 0\}. \tag{4}$$

Here, the sub-zero level set of auxiliary value $V$ becomes the epigraph of the constrained value $v$. The proof is listed in Appendix A.1.

**Lemma 3.2** (Optimality Condition). For all $(t, x, z) \in [0, \infty) \times \mathcal{X} \times \mathbb{R}$, consider a small enough $h > 0$, the auxiliary value function $V$ satisfies

$$V(x, z) = \min_{u \in \mathcal{U}} \max \left\{ \max_{\tau \in [t, t+h]} c(x(\tau)), \gamma^h V(x(t + h), z(t + h)) \right\}. \tag{5}$$

The proof is listed in Appendix A.2.

**Theorem 3.3** (Epigraph-based HJB PDE). Let $V : \mathcal{X} \times \mathbb{R} \to \mathbb{R}$ be the auxiliary value function defined in Eq. 3. Then $V$ is the unique viscosity solution of the following HJB PDE for all $(t, x, z) \in [0, \infty) \times \mathcal{X} \times \mathbb{R}$

$$\max \left\{ \max_{\tau \in [t, \infty]} c(x) - V(x, z), \ \min_{u \in \mathcal{U}} \mathcal{H}(x, z, \nabla_x V, \partial_z V) \right\} = 0, \tag{6}$$

where $\mathcal{H}(x, z, \nabla_x V, \partial_z V)$ is Hamiltonian and satisfies $\mathcal{H} = \nabla_x V \cdot f(x, u) - \partial_z V \cdot l(x, u) + \ln \gamma \cdot V$ and optimal control $u^* = \arg\min_{u \in \mathcal{U}} \mathcal{H}$. The derivation proof is provided in Appendix A.3.

## 3.2 EPIGRAPH LEARNING FRAMEWORK

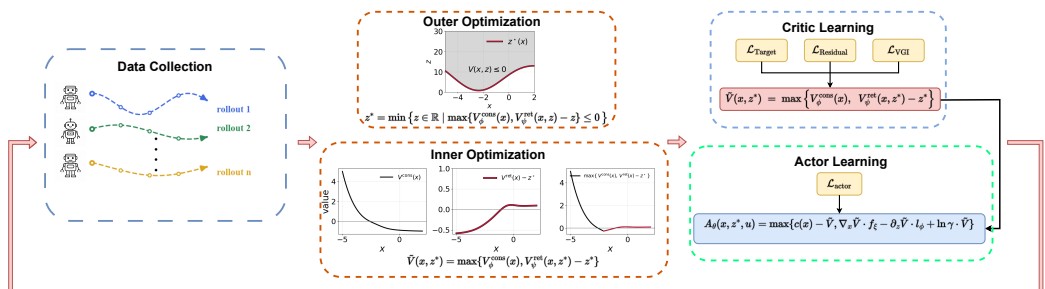

Figure 1: Overview of the proposed epigraph-based CT-MARL framework. The pipeline begins with data collection, where individual agent rollouts are aggregated into a centralized rollout $\mathcal{X}_R$ for the training; the outer optimization computes optimal $z^*$ to balance discounted cumulative cost and safety constraints; the inner optimization corresponds to critic learning, where return networks $V_\psi^{\text{ret}}(x)$ and constraint value networks $V_\phi^{\text{cons}}(x)$ are optimized jointly with the optimal auxiliary state $z^*$; and actor learning leverages the advantage function to improve policies.

As illustrated in Fig. 1, our framework integrates the epigraph-based inner-outer optimization (Zhang et al., 2025b) into the actor-critic paradigm. The outer loop updates $z^*$ along the rollout by solving Eq. 7, ensuring that the critic is trained with the minimal $z$ that simultaneously satisfies both costs and safety constraints.

$$\min_{z \in \mathbb{R}} z \quad \text{s.t.} \quad \min_\pi \max \left\{ \sup_{\tau \ge t} c(x(\tau)), \int_t^\infty \gamma^{\tau - t} l(x(\tau), \pi(\tau)) \, d\tau - z \right\} \le 0. \tag{7}$$

In the inner loop, the critic is trained as follows: the return and constraint value networks ($V_\psi^{\text{ret}}(x)$ and $V_\phi^{\text{cons}}(x)$) are optimized using $z^*$ to approximate the auxiliary value function $\tilde{V}(x, z^*)$. This stabilized critic subsequently supplies the learning signals for decentralized actors, which map local observations to continuous-time policies under the standard centralized training decentralized execution setup (Foerster et al., 2018; Lowe et al., 2017). We next describe the revised outer optimization in detail, focusing on solving the optimal auxiliary state $z^*$ that trades off discounted cost against safety violations without costly root-finding algorithms (So & Fan, 2023; So et al., 2024; Zhang et al., 2025b).

### 3.2.1 REVISED OUTER OPTIMIZATION

We seek the minimal $z$ such that the epigraph-based value $V$ remains non-positive, as defined in Eq. 4. Using the return and constraint value network learned by the critic, the optimal auxiliary state $z^*$ can be found by solving for the minimal feasible solution

$$z^* = \min \left\{ z \in \mathbb{R} \mid \max\{V_\phi^{\text{cons}}(x), V_\psi^{\text{ret}}(x) - z\} \leq 0 \right\}, \tag{8}$$

where return value network $V_\psi^{\text{ret}}(x)$ that approximates the discounted cumulative cost $\int_t^\infty \gamma^{\tau-t} l(x(\tau), \pi(\tau)) \, d\tau$, and constraint value network $V_\phi^{\text{cons}}(x)$ represents the violation for worst-case future constraints $\sup_{\tau \geq t} c(x(\tau))$.

In previous epigraph formulations (Tayal et al., 2025; Zhang et al., 2025b), the outer problem is solved during the execution phase: $z$ is sampled along the rollouts during training, and $z^*$ is computed at execution time via root-finding (Stoer et al., 1980). This design has two drawbacks in CT-MARL: (1) the random sampling of $z$ introduces nonstationary noise that destabilizes the updates of actor and critic and further leads to poor convergence; (2) at execution, root-finding must be performed at every step, which is computationally expensive and often incompatible with real-time requirements. In contrast, we design the return and constraint value networks as functions of the states $x$ solely. We then integrate the outer optimization into actor-critic training: for each episode, $z^*$ is computed using the current learned value $\tilde{V}$ along the predicted rollout. The actor is then trained against a $z$-independent critic, yielding a $z$-independent policy $\pi(x)$. This design ensures stable actor training, and enables real-time deployment by eliminating the need for root-finding during execution. Since the critic's value networks are $z$-independent, the outer optimization is simplified to a scalar search for $z^*$, which adds negligible cost to model training.

### 3.2.2 INNER OPTIMIZATION WITH CRITIC LEARNING

The inner optimization is responsible for updating the PINN-based critic networks. Given a task-dependent range $[z_{\min}, z_{\max}]$, the outer optimization computes $z^*$, which is then clipped to this range (i.e., $z^* \leftarrow \min\{\max\{z^*, z_{\min}\}, z_{\max}\}$) before being used to train the critic module. The critic consists of two value networks: a return value network $V_\psi^{\text{ret}}(x)$, and a constraint value network $V_\phi^{\text{cons}}(x)$. Together with the computed $z^*$ from Eq. 8, these define the composite epigraph-based value function

$$\tilde{V}(x, z^*) = \max \left\{ V_\phi^{\text{cons}}(x), V_\psi^{\text{ret}}(x) - z^* \right\}. \tag{9}$$

To ensure stable and accurate training, we employ three complementary losses.

**(i) Residual Loss.** We use PINN architecture (Mukherjee & Liu) to approximate the value function governed by epigraph-based HJB PDEs, and introduce a residual loss that penalizes violations of the corresponding PDEs

$$\mathcal{L}_{\text{Residual}} = \left\| \max \left\{ c(x) - \tilde{V}, \min_{u \in \mathcal{U}} \left[ \nabla_x \tilde{V} \cdot f(x, u) - \partial_z \tilde{V} \cdot l(x, u) + \ln \gamma \cdot \tilde{V} \right] \right\} \right\|_2^2. \tag{10}$$

**(ii) Target Loss.** In standard PINNs, a boundary loss is combined with the PDE residual to approximate PDE solutions (Cai et al., 2021; Raissi et al., 2019). In the infinite-horizon setting, however, no boundary condition is available, and training the critic only on residuals is insufficient: optimization may converge, but to incorrect PDE solutions (Wang et al., 2022). To address this, we add a rollout-based *target loss* that measures the discrepancy between the epigraph-based value approximation with a numerical target defined by Eq. 3. For each episode, the current value $\tilde{V}$ generates a closed-loop trajectory $\{x(\tau), u(\tau)\}_{\tau=t}^\infty$; from this trajectory we construct the target $V_{\text{tgt}}(x, z) = \max \left\{ \max_{\tau \in [t, \infty]} c(x(\tau)), \int_t^\infty \gamma^{\tau-t} l(x(\tau), u(\tau)) \, d\tau - z^* \right\}$ and minimize the squared error

$$\mathcal{L}_{\text{Target}} = \left\| V_{\text{tgt}}(x, z^*) - \max\{ V_\phi^{\text{cons}}(x), V_\psi^{\text{ret}}(x) - z^* \} \right\|_2^2. \tag{11}$$

**(iii) Value Gradient Iterations.** Standard PINN training in multi-agent settings often struggles to approximate accurate value functions, primarily because the learned value gradients are inaccurate or unstable (Wang et al., 2025; Zhang et al., 2024). The VGI techniques (Eberhard et al., 2025; Wang et al., 2025) are designed to enhance the quality of learned value gradients. In our framework, accurate gradients $\nabla_x V(x)$ are crucial for precise value approximations, which in turn affect

actor learning and ultimately determine the quality of the resulting policies. To establish the theoretical basis of this module, we follow Theorem 3.4 in Bokanowski et al. (2021) and Theorem 2 in Hermosilla & Zidani (2023). Here, $u_t$ is the optimal control input in this paper.

$$\nabla_{x_t}\tilde{V}(x_t) = \nabla_{x_t}(\chi(x_t)l(x_t, u_t) + (1 - \chi(x_t))c(x_t))\Delta t + \gamma^{\Delta t}\nabla_{x_{t+\Delta t}}\tilde{V}(x_{t+\Delta t}) \cdot \nabla_{x_t}f(x_t, u_t),$$
(12)

where the $\chi(x_t) := \mathbf{1}\{V_\psi^{\text{ret}}(x_t) - z_t \geq V_\phi^{\text{cons}}(x_t)\}$. As shown in Eq. 12, the value gradient satisfies a recursive relation coupling the local cost gradient with the backpropagated dynamics term. The overall critic objective is a weighted sum of the three losses as

$$\mathcal{L}_{\text{Critic}} = \lambda_{\text{res}}\mathcal{L}_{\text{Residual}} + \lambda_{\text{tgt}}\mathcal{L}_{\text{Target}} + \lambda_{\text{vgi}}\mathcal{L}_{\text{VGI}},$$
(13)

where the weights $(\lambda_{\text{res}}, \lambda_{\text{tgt}}, \lambda_{\text{vgi}})$ are selected to keep the losses on comparable scales and are determined via grid search.

### 3.2.3 ACTOR LEARNING

After introducing the inner-outer optimization for critic learning, we turn to the actor learning. We first define the epigraph-based Q-function, which is used for deriving policy update rules.

**Definition 2** (Epigraph-based Q-function). Following the definition in (So & Fan, 2023), for any state-action pair $(x_t, u_t)$ and auxiliary state $z_t$, the epigraph-based Q-function is defined

$$Q(x_t, z_t^*, u_t) = \max\left\{c(x_t), \gamma^h V(x_{t+h}, z_{t+h}^*)\right\}.$$
(14)

where $x_{t+h}$ and $z_{t+h}^*$ are the states and optimal auxiliary state at $t + h$, respectively. $h$ is a short time interval.

**Lemma 3.4** (Epigraph-based advantage function). The epigraph-based advantage function

$$A(x_t, z_t^*, u_t) = Q(x_t, z_t^*, u_t) - V(x_t, z_t^*)$$
(15)

is equivalent to epigraph-based HJB PDE when $h \to 0$

$$A(x_t, z_t^*, u_t) = \max\{c(x_t) - V(x_t, z_t^*), \nabla_{x_t}V \cdot f(x_t, u_t) - \partial_{z_t}V \cdot l(x_t, u_t) + \ln\gamma \cdot V\}. \quad (16)$$

In practice, evaluating the epigraph-based advantage in Eq. 16 requires knowledge of the true dynamics $f(x, u)$ and cost function $l(x, u)$. Since these quantities are generally unknown in model-free reinforcement learning, we replace them with neural networks that are jointly trained alongside the actor. The derivation of the epigraph-based advantage function is listed at Appendix A.4.

**Dynamics and Cost Networks.** To assist with the policy training, we employ two neural networks: a dynamics network $f_\xi(x, u, \Delta_t)$ that predicts the next state $x'$ given the current state–action pair, and a cost network $l_\phi(x, u, \Delta_t)$ that estimates the instantaneous stage cost. Both models are trained via supervised regression using observed transitions $(x, u, x', l)$ from the environment. Specifically, the training losses are

$$\mathcal{L}_{\text{dyn}}(\xi) = \left\|f_\xi(x, u, \Delta_t) - x'\right\|_2^2, \qquad \mathcal{L}_{\text{cost}}(\phi) = \left\|l_\phi(x, u, \Delta_t) - l(x, u)\right\|_2^2, \qquad (17)$$

where $x'$ is the observed next state and $l(x, u)$ is the empirical cost signal. Equivalently, the dynamics learning can be interpreted as approximating the continuous-time derivative dynamics $(f_\xi(x, u, \Delta_t) - x)/\Delta t$.

**Actor Update with Learned Models.** By substituting $\tilde{V}(x, z^*)$, $f_\xi$ and $l_\phi$ into the epigraph advantage expression Eq. 16, we obtain a differentiable surrogate

$$A_\theta(x, z^*, u) = \max\{c(x) - \tilde{V}, \nabla_x\tilde{V} \cdot f_\xi - \partial_z\tilde{V} \cdot l_\phi + \ln\gamma \cdot \tilde{V}\}.$$
(18)

The actor $\pi_\theta(u \mid x)$ is updated by minimizing the expected surrogate advantage

$$\mathcal{L}_{\text{actor}}(\theta) = \mathbb{E}_{x\sim\mathcal{X}_R, u\sim\pi_\theta(\cdot|x)}\left[A_\theta(x, z^*, u)\right],$$
(19)

where $\mathcal{X}_R$ is the sampled data along the rollout.

Specifically, we adopt a centralized-training decentralized-execution structure: each agent's actor $\pi_i(o_i, \Delta_t)$ takes its local observation $o_i$ as input, while the training signal is derived from the state $x$. The overall training pipeline is summarized in Algorithm 1 in Appendix B.

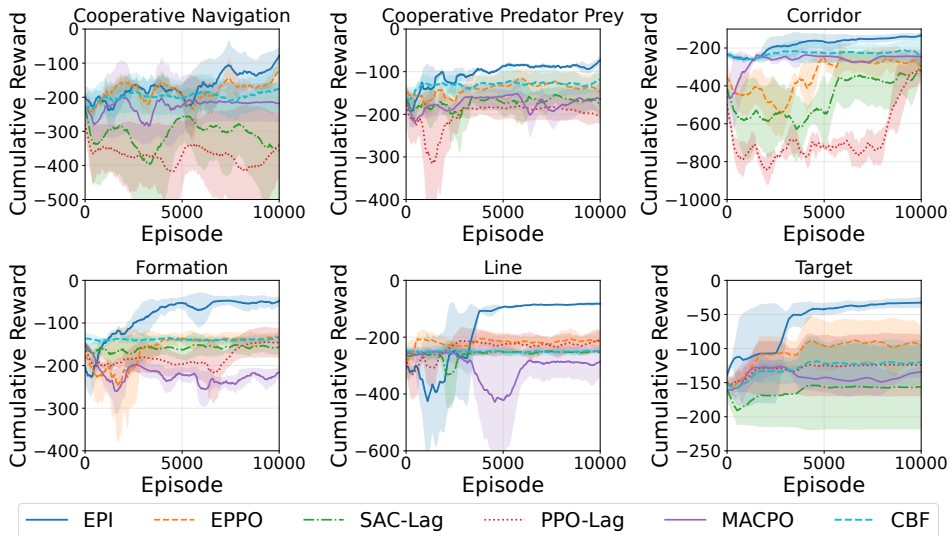

Figure 2: Overall results for adapted MPE environments.

# 4 EXPERIMENTAL RESULTS

We organize our empirical study around the following research questions: **Q1.** How well does our method balance discounted cumulative cost and constraint satisfaction compared to state-of-the-art baselines? **Q2.** How does the different loss component in critic learning contribute to stable training and accurate value approximations? **Q3.** How does performance change when training with versus without the epigraph reformulation? **Q4.** How sensitive is the epigraph formulation to the choice of the auxiliary variable $z$ during training? **Q5.** How robust is the method under stochastic disturbances, and how does performance degrade under model-mismatch noise? **Q6.** How does the performance change under different discretization resolutions $\Delta t$?

## 4.1 BENCHMARKS AND BASELINES.

To evaluate our approach under continuous-time environments with safety constraints, we consider two adapted benchmarks: the safe continuous-time MPE (Lowe et al., 2017; Wang et al., 2025) and continuous-time Safe MA-MuJoCo (Gu et al., 2023; Wang et al., 2025). In MPE, we design several scenarios including *Corridor*, *Formation*, *Line*, *Target*, *Simple Spread*, and *Cooperative Predator–Prey*. These tasks typically place agents in environments with obstacles and require them to avoid both collisions with obstacles and collisions with other agents while navigating or pursuing their objectives. In MuJoCo, we adapt several scenarios such as Half Cheetah and Ant into continuous-time versions and introduce randomly placed walls as obstacles. The agents must coordinate to move forward efficiently while avoiding crashing into walls, ensuring that the learned policies account for both locomotion and safety considerations.

Lastly, we design a didactic example based on a constrained coupled oscillator, which admits an analytical ground-truth solution for both value functions and actions. This example provides a transparent testbed to directly validate the correctness of our learned critics against exact solutions. Full details of the agent setups, metrics,

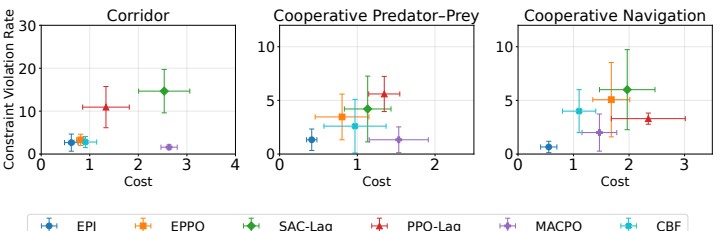

Figure 3: Performance of constraints and cost over MPE settings.

state and action spaces, and cost specifications are provided in the Appendix C. We compare our approach EPI with MACPO (Gu et al., 2021), MAPPO-Lag (Gu et al., 2021), SAC-Lag (Haarnoja et al., 2018), EPPO (Zhang et al., 2025b) and CBF (Zhang et al., 2025a). The first three represent

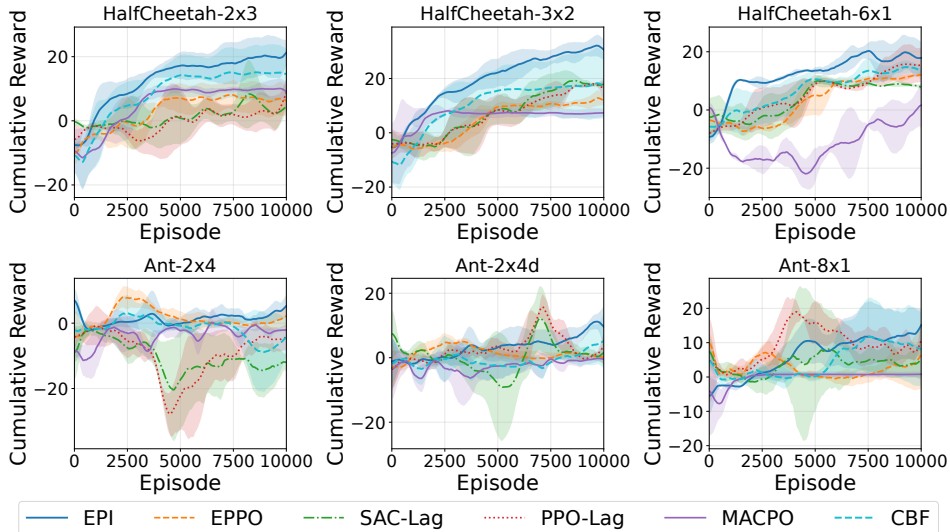

Figure 4: Overall results for adapted multi-agent MuJoCo environments.

the most widely used families of safe MARL algorithms: trust-region based methods (MACPO) and Lagrangian based methods (MAPPO-Lag, SAC-Lag), covering both on-policy and off-policy learning. We also include EPPO as an epigraph-based baseline that follows the traditional epigraph optimization framework. We additionally include a control barrier function (CBF) baseline, which enforces safety through model-based barrier certificates and is commonly used in safe multi-agent control. Although these algorithms were originally developed in the discrete-time setting, we adapt them to continuous time by equipping their critics with the same PDE residual loss used in our method. Since the performance gap between discrete-time and continuous-time algorithms has already been well studied (Tallec et al., 2019; De Asis & Sutton, 2024), our baselines focus only on isolating the effect of different safety mechanisms (trust-region, Lagrangian, or epigraph).

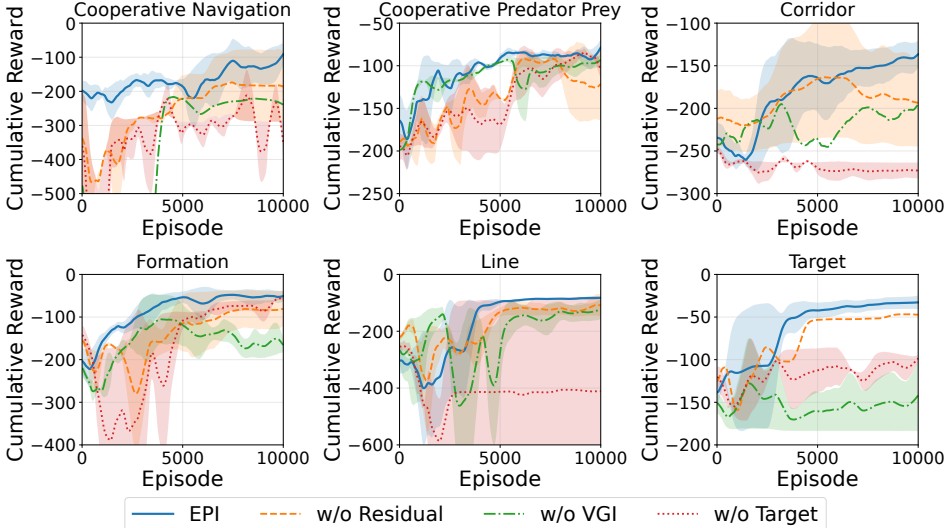

Figure 5: Ablation study of different loss terms in critic network over MPE.

## 4.2 RESULTS ANALYSIS

In this section, we present a systematic analysis of the results, addressing each research question in turn. **Q1.** Our method consistently outperforms all baselines across both adapted MPE and MuJoCo environments in Fig. 2 and Fig. 4. We adopt the same reward design commonly used in prior safe MARL works such as MACPO (Gu et al., 2021). Specifically, the reward is the combination of the task cost provided by the environment (e.g., distance to the target

in MPE) and the safety penalty provided by the environment (e.g., collision penalty between agents or with obstacles), as detailed in Appendix C, which directly reflects performance under both objectives. In Fig. 3 and 6, each point corresponds to the average performance of one algorithm, with horizontal and vertical bars denoting standard deviations. Since the goal is to minimize both cost and constraint violations, the lower-left corner of each panel represents the desirable region. These results show that our algorithm EPI achieves nearly the lowest cost and constraint violation in every scenarios. Specifically, EPPO often remains stuck at suboptimal solutions because it randomly samples the auxiliary state $z$ instead of using $z^*$ for model training, introducing noise that disrupts policy updates and prevents stable convergence.

MACPO enforces constraints through a hard trust-region style update, which yields strong violation rejection but tends to be overly conservative. SAC-Lag and MAPPO-Lag rely on Lagrangian relaxation, which is known to suffer from instability when balancing objectives under tight safety requirements (Zhang et al., 2025b).

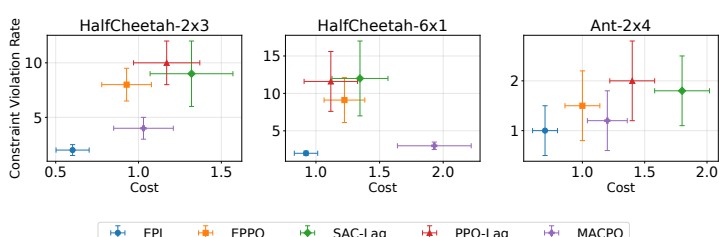

Figure 6: Performance of constraints and cost over MuJoCo settings.

CBF achieves reasonable constraint-violation levels but tends to be conservative. The CBF condition relies on the gradient of a learned barrier function $\nabla B(x)$, approximation errors in this component can distort the effective safe set and degrade the overall performance.

**Q2.** The ablation results in Fig. 5 clearly demonstrate the importance of each loss component in critic learning. It presents the cumulative reward performance of our full method compared with its ablation variants across representative continuous-time MPE tasks.

Removing the target loss or the VGI loss significantly degrades performance, whereas removing the residual loss has only a minor effect. This difference stems from the fact that, unlike existing HJ-based PINN methods (Zhang et al., 2024; Tayal et al., 2025; Cai et al., 2021) that address finite-horizon problems with boundary conditions, our framework targets the infinite-horizon setting where no such boundary conditions are available. In this case, the target loss serves as an anchor to stabilize value approximations, ensuring that value function $V(x)$ does not drift arbitrarily, while the VGI loss enforces consistency of the learned value gra-

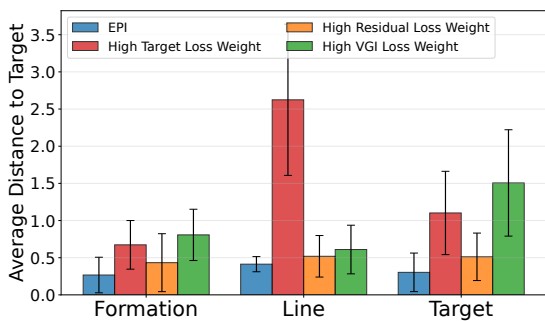

Figure 7: Weighted loss performance.

dients, which are crucial for both accurate value approximations and policy improvement. In contrast, the HJB residual loss mainly regularizes the PDE structure, but its role becomes less critical once the value gradients are optimized by VGI. As a result, the removal of VGI has a severe impact, since inaccurate value gradients directly harm both critic accuracy and actor updates, while the residual loss contributes less critically to overall training stability.

The grouped bars in Fig. 7 report the average distance to the target (lower is better) for three MPE tasks (Formation, Line, and Target) under different loss weightings. The balanced setting (EPI) attains the smallest distance in all tasks and shows the tightest variability. Over-emphasizing any single component degrades performance: increasing the target loss weight is particularly harmful on Line (large increase in distance), while overweighting ($\times 20$) the residual or the VGI loss also worsens results relative to EPI, though to a lesser extent. These ablations support using the balanced weighting adopted by EPI.

**Q3.** We generate one trajectory using EPI and collect the visited states. On these same states, we compare the value and policy from three methods: EPI, Ground Truth via the LQR method (details

in the Appendix C), and an ablation without the epigraph reformulation, where the state constraint is treated as a collision penalty added to the cost function $l$, making the value function discontinuous.

For Ground Truth, the value is computed as the discounted cumulative cost. While for the EPI and ablation without the epigraph form, the value is predicted through the trained value network. EPI closely

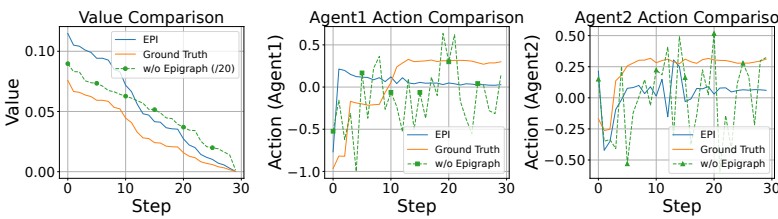

Figure 8: Performance with and without epigraph reformulation.

tracks the Ground Truth in both value and actions for both agents, indicating accurate value approximation and stable control policies. In contrast, the ablation without the epigraph form exhibits severely mis-scaled value predictions (we plot it after a $\times\frac{1}{20}$ scaling to share the same y-axis) and noticeably unstable actions, which in practice are more likely to violate constraints because the discontinuous value function is not addressed by the epigraph form. The poor performance of the ablation without epigraph stems from the discontinuity of the value function when state constraints are directly encoded as hard penalties. Such discontinuities are notoriously difficult to approximate with neural networks, leading to severely mis-scaled value predictions and unstable gradients for policy updates. By contrast, the epigraph reformulation converts the discontinuous penalty into a continuous and smooth upper-bound optimization, which stabilizes critic learning and yields reliable policies.

**Q4.** To better understand how model performance depends on $z$, we test two MPE tasks (Formation and Line) under different values of $z$. Specifically, we train the EPI model with $z \in \{z^* - 0.5z_{\max}, z^* - 0.2z_{\max}, z^*, z^* + 0.2z_{\max}, z^* + 0.5z_{\max}\}$. Fig. 9 reports the results, where the $x$-axis indicates cost and the $y$-axis denotes the constraint violation rate.

Compared with the optimal auxiliary state $z^*$, using a suboptimal $z$ shifts the trade-off between cost and constraint satisfaction, often resulting in either much higher violation rates or larger costs. Specifically, a smaller $z$ (e.g., $z^* - 0.2z_{\max}, z^* - 0.5z_{\max}$) significantly increases the violation rate while only

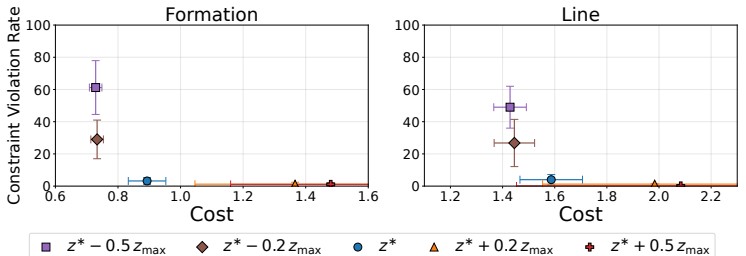

Figure 9: Sensitivity test of different z choices.

slightly reducing cost. Getting back to the epigraph form $\max\{V_\phi^{\mathrm{cons}}(x), \ V_\psi^{\mathrm{ret}}(x,z) - z\}$, a smaller $z$ makes $V^{\mathrm{ret}}(x,z) - z$ lager than $V_\phi^{\mathrm{cons}}(x)$, so the return term dominates in the epigraph form. As a result, the optimization prioritizes reward improvement while neglecting constraint satisfaction, leading to frequent violations. In contrast, when $z$ is larger than $z^*$ (e.g., $z^* + 0.2z_{\max}$, $z^* + 0.5z_{\max}$), the term $V^{\mathrm{ret}}(x,z) - z$ becomes smaller than $V_\phi^{\mathrm{cons}}(x)$, making constraint value dominate in the epigraph form. This forces the critic and actor to emphasize constraint satisfaction, which reduces violations but increases cost.

## 5 CONCLUSION

In this paper, we propose an epigraph-based framework for CT-MARL that addresses the challenges of balancing reward maximization with constraint satisfaction. By reformulating the problem through the epigraph forms, we introduced an inner–outer optimization procedure that enables stable critic learning and effective policy updates. Our design further integrates different losses in critic learning, including target, residual, and VGI losses, to anchor value approximations and improve gradient accuracy in the infinite-horizon setting. Through extensive experiments in both adapted MPE and MuJoCo benchmarks, we demonstrated that our method consistently outperforms state-of-the-art baselines in terms of both cost reduction and constraint satisfaction.

## ACKNOWLEDGEMENT

This material is based upon work supported by the Air Force Office of Scientific Research under award number FA9550-24-1-0233. Any opinions, findings, and conclusions or recommendations expressed in this material are those of the author(s) and do not necessarily reflect the views of the United States Air Force.

## ETHICS STATEMENT

This work focuses on decision-making for the continuous-time constrained MDP problems. All experiments are conducted entirely in simulation and do not involve human subjects or personal data.

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

## A  MATHEMATICAL PROOF

### A.1  LEMMA 3.1: EQUIVALENCE OF TWO VALUE FUNCTIONS

*Proof.* Following proofs in (Lee, 2022; Zhang et al., 2024)), Eq. 4 implies the following equivalence

$$v(x) - z \le 0 \quad \Longleftrightarrow \quad V(x, z) \le 0$$

To prove the above relation, we first start from $v(x) - z \le 0$, which implies that there exists a joint control input $u \in \mathcal{U}$ such that

$$\int_t^\infty \gamma^{\tau-t} l(x(\tau), u(\tau)) d\tau - z \le 0,$$

with $c(x(\tau)) \le 0$ for $\forall \tau \ge t$. Thus, there will exist a joint control $u$ such that $V(x, z) \le 0$.

Second, when $V(x, z) \le 0$ and $c(x(\tau)) \le 0$ for $\forall \tau \ge t$ hold, it implies that there exists $u \in \mathcal{U}$ such that

$$\int_t^\infty \gamma^{\tau-t} l(x(\tau), u(\tau)) d\tau - z \le 0,$$

which concludes $v(x) - z \le 0$. Therefore, the Lemma 3.1 is proved. □

### A.2  LEMMA 3.2: OPTIMALITY CONDITION

*Proof.* Following proofs in (Lee, 2022; Zhang et al., 2024; Evans, 2022), given all $(t, x, z) \in [0, \infty) \times \mathcal{X} \times \mathbb{R}$ and select a enough small $h > 0$. There exist two different joint control inputs $(u_1(\cdot), u_2(\cdot)) \in \mathcal{U}$ such that

$$u(\tau) = \begin{cases} u_1(\tau), & \tau \in [t, t+h], \\ u_2(\tau), & \tau \in (t+h, \infty). \end{cases}$$

Then we have the following transformation for Eq. 3

$$V(x, z) = \min_{u_1 \in \mathcal{U}, u_2 \in \mathcal{U}} \max \Big\{ \max_{\tau \in [t, t+h]} c(x(\tau)), \max_{\tau \in [t+h, \infty)} c(x(\tau)),$$

$$\int_t^\infty \gamma^{\tau-t} l(x(\tau), u(\tau)) d\tau - z(t) \Big\}$$

$$= \min_{u_1 \in \mathcal{U}} \max \Big\{ \max_{\tau \in [t, t+h]} c(x(\tau)), \min_{u_2 \in \mathcal{U}} \max \Big\{ \max_{\tau \in [t+h, \infty)} c(x(\tau)), \int_t^{t+h} \gamma^{\tau-t} l(x(\tau), u(\tau)) d\tau$$

$$+ \int_{t+h}^\infty \gamma^{\tau-t} l(x(\tau), u(\tau)) d\tau - \big(z(t+h) + \int_t^{t+h} \gamma^{\tau-t} l(x(\tau), u(\tau)) d\tau\big) \Big\} \Big\}$$

$$= \min_{u_1 \in \mathcal{U}} \max \Big\{ \max_{\tau \in [t, t+h]} c(x(\tau)), \min_{u_2 \in \mathcal{U}} \max \Big\{ \max_{\tau \in [t+h, \infty)} c(x(\tau)),$$

$$\int_{t+h}^\infty \gamma^{\tau-t} l(x(\tau), u(\tau)) \, d\tau - z(t+h) \Big\} \Big\}$$

$$\approx \min_{u_1 \in \mathcal{U}} \max \Big\{ \max_{\tau \in [t, t+h]} c(x(\tau)), \min_{u_2 \in \mathcal{U}} \max \Big\{ \max_{\tau \in [t+h, \infty)} c(x(\tau)),$$

$$\gamma^h \big( \int_{t+h}^\infty \gamma^{\tau-(t+h)} l(x(\tau), u(\tau)) \, d\tau - z(t+h) \big) \Big\} \Big\}$$

$$= \min_{u_1 \in \mathcal{U}} \max \Big\{ \max_{\tau \in [t, t+h]} c(x(\tau)), \gamma^h V(x(t+h), z(t+h)) \Big\}$$

$$= \min_{u \in \mathcal{U}} \max \Big\{ \max_{\tau \in [t, t+h]} c(x(\tau)), \gamma^h V(x(t+h), z(t+h)) \Big\}$$

Therefore, the Lemma 3.2 is proved. □

## A.3 THEOREM 3.3: EPIGRAPH-BASED HJB PDE

*Proof.* Following proofs in (Lee, 2022; Zhang et al., 2024; Evans, 2022), given all all $(t, x, z) \in [0, \infty) \times \mathcal{X} \times \mathbb{R}$ with a small horizon $\Delta t > 0$, we apply Lemma 3.2 and Taylor expansion to derive the epigraph-based HJB PDE as follows

$$V(x, z) = \min_{u \in \mathcal{U}} \max \Big\{ \max_{\tau \in [t, t + \Delta t]} c(x(\tau)), \, \gamma^h V(x(t + \Delta t), z(t + \Delta t)) \Big\}$$

$$\approx \min_{u \in \mathcal{U}} \max \Big\{ c(x), (1 + \ln \gamma \Delta t)(V(x, z) + \nabla_x V \cdot f(x, u) \Delta t - \partial_z V \cdot l(x, u) \Delta t + o(\Delta t)) \Big\}$$

$$= \max \Big\{ c(x), (1 + \ln \gamma \Delta t) \min_{u \in \mathcal{U}} (V(x, z) + \nabla_x V \cdot f(x, u) \Delta t - \partial_z V \cdot l(x, u) \Delta t + o(\Delta t)) \Big\}$$

Subtracting $V(x, z)$ from both sides of above equality, dividing by $\Delta t$, and letting $\Delta t \to 0$ yields the following HJB PDE, where $V(x, z)$ is the optimal solution to such PDE.

$$\max \Big\{ c(x) - V(x, z), \min_{u \in \mathcal{U}} \big[ \nabla_x V \cdot f(x, u) - \partial_z V \cdot l(x, u) + \ln \gamma \cdot V \big] \Big\} = 0.$$

Here $\mathcal{H} = \nabla_x V \cdot f(x, u) - \partial_z V \cdot l(x, u) + \ln \gamma \cdot V$ is Hamiltonian and optimal control $u^* = \arg\min_{u \in \mathcal{U}} \mathcal{H}$.

Next we prove that $V(x, z)$ is the unique viscosity solution to the epigraph-based HJB PDE using the contradiction technique. First, for $U \in C^\infty(\mathcal{X} \times \mathbb{R})$ such that $V - U$ has local maximum at $(x_0, z_0) \in \mathcal{X} \times \mathbb{R}$ and $(V - U)(x_0, z_0) = 0$, we will prove

$$\max \Big\{ c(x_0) - U(x_0, z_0), \min_{u \in \mathcal{U}} \big[ \nabla_x U(x_0, z_0) \cdot f(x_0, u) - \partial_z U(x_0, z_0) \cdot l(x_0, u) + \ln \gamma \cdot U(x_0, z_0) \big] \Big\} \geq 0.$$

Suppose the above inequality is not correct. We consider that there exists $\theta > 0$ and $\tilde{u} \in \mathcal{U}$ such that

$$c(x) - U(x_0, z_0) \leq -\theta,$$
$$\nabla_x U \cdot f(x, \tilde{u}) - \partial_z U \cdot l(x, \tilde{u}) + \ln \gamma \cdot U \leq -\theta.$$

for all points $(x, z)$ sufficiently close to $(x_0, z_0)$: $\|x(s) - x_0\| + |z(s) - z_0| < h$ for small enough $h > 0$, where $s \in [t_0, t_0 + h]$. Under the assumptions in Sec. 3.1.1, and given state trajectories $x$ and $z$ evolved from the initial conditions $x = x_0$ and $z = z_0$ according to the corresponding dynamics, the following inequality holds

$$c(x(s)) - U(x_0, z_0) \leq -\theta,$$
$$\nabla_x U(x(s), z(s)) \cdot f(x(s), \tilde{u}) - \partial_z U(x(s), z(s)) \cdot l(x(s), \tilde{u}) + \ln \gamma \cdot U(x(s), z(s)) \leq -\theta.$$

Since $V - U$ has a local maximum at $(x_0, z_0)$, we can have that

$$\min_{u \in \mathcal{U}} \big[ \gamma^h V(x(t_0 + h), z(t_0 + h)) - V(x_0, z_0) \big]$$
$$\leq \min_{u \in \mathcal{U}} \big[ \gamma^h U(x(t_0 + h), z(t_0 + h)) - U(x_0, z_0) \big]$$
$$= \min_{u \in \mathcal{U}} \big[ (\nabla_x U(x(t_0), z(t_0)) \cdot f(x(t_0), u) - \partial_z U(x(t_0), z(t_0)) \cdot l(x(t_0), u) + \ln \gamma \cdot U(x(t_0), z(t_0)))h \big]$$
$$\leq -\theta h$$

We know that Lemma 2 implies

$$V(x_0, z_0) = \min_{u \in \mathcal{U}} \max \Big\{ \max_{s\tau \in [t_0, t_0 + h]} c(x(s)), \, \gamma^h V(x(t_0 + h), z(t_0 + h)) \Big\}.$$

By subtracting $U(x_0, z_0)$ on both side, we have

$$(V - U)(x_0, z_0) = \min_{u \in \mathcal{U}} \max \Big\{ c(x(s)) - U(x_0, z_0), \, \gamma^h V(x(t_0 + h), z(t_0 + h)) - U(x_0, z_0) \Big\}.$$

Since $(V - U)(x_0, z_0) = 0$ holds such that $V(x_0, z_0) = U(x_0, z_0)$, then we will have that

$$\min_{u \in \mathcal{U}} \max \Big\{ c(x(s)) - V(x_0, z_0), \, \gamma^h V(x(t_0 + h), z(t_0 + h)) - V(x_0, z_0) \Big\} = \min_{u \in \mathcal{U}} \max \{\theta, \theta h\} > 0,$$

which has a contradiction with $(V - U)(x_0, z_0) = 0$. Thus we prove that

$$\max\Big\{c(x_0)-U(x_0, z_0), \ \min_{u\in\mathcal{U}}\big[\nabla_x U(x_0, z_0)\cdot f(x_0, u_0)-\partial_z U(x_0, z_0)\cdot l(x_0, u_0)+\ln\gamma\cdot U(x_0, z_0)\big]\Big\} \geq 0.$$

Second, for $U \in C^\infty(\mathcal{X} \times \mathbb{R})$ such that $V - U$ has local minimum at $(x_0, z_0) \in \mathcal{X} \times \mathbb{R}$ and $(V - U)(x_0, z_0) = 0$, we will prove

$$\max\Big\{c(x_0)-U(x_0, z_0), \ \min_{u\in\mathcal{U}}\big[\nabla_x U(x_0, z_0)\cdot f(x_0, u_0)-\partial_z U(x_0, z_0)\cdot l(x_0, u_0)+\ln\gamma\cdot U(x_0, z_0)\big]\Big\} \leq 0.$$

The definition of auxiliary value $V(x, z)$ shows that

$$V(x, z) = \min_{u\in\mathcal{U}} \max\left\{ \max_{\tau\in[t,\infty]} c(x(\tau)), \int_t^\infty \gamma^{\tau-t}l(x(\tau), u(\tau))d\tau - z\right\}$$

$$\geq \min_{u\in\mathcal{U}} \max\left\{ c(x_0), \int_t^\infty \gamma^{\tau-t}l(x(\tau), u(\tau))d\tau - z\right\}$$

for all $u \in \mathcal{U}$. By subtracting $U(x_0, z_0)$ on both sides, we have

$$0 = (V - U)(x_0, z_0) \geq \max\{c(x_0) - U(x_0, z_0), \int_t^\infty \gamma^{\tau-t}l(x, u)d\tau - z_0 - U(x_0, z_0)\}.$$

The rest of the proof is to show

$$\min_{u\in\mathcal{U}}\big[\nabla_x U(x_0, z_0) \cdot f(x_0, u) - \partial_z U(x_0, z_0) \cdot l(x_0, u) + \ln\gamma \cdot U(x_0, z_0)\big] \leq 0.$$

Suppose the above inequality is not correct. We consider that there exists $\theta > 0$ such that

$$\min_{u\in\mathcal{U}}\big[\nabla_x U(x, z) \cdot f(x, u) - \partial_z U(x, z) \cdot l(x, u) + \ln\gamma \cdot U(x, z)\big] \geq \theta,$$

for all points $(x, z)$ sufficiently close to $(x_0, z_0)$: $\|x - x_0\| + |z - z_0| < h$ for small enough $h > 0$, where $s \in [t_0, t_0 + h]$. Given state trajectories $x$ and $z$ that evolve from the initial conditions $x = x_0$ and $z = z_0$ under the corresponding dynamics with any control $\tilde{u} \in \mathcal{U}$, where

$$\tilde{u}(s) = \arg\min_{\tilde{u}\in\mathcal{U}} \big\{\nabla_x U(x(s), z(s)) \cdot f(x(s), \tilde{u}) - \partial_z U(x(s), z(s)) \cdot l(x(s), \tilde{u})$$

$$+ \ln\gamma \cdot U(x(s), z(s))\big\}.$$

Then we have the following condition that holds

$$\nabla_x U(x(s), z(s)) \cdot f(x(s), \tilde{u}) - \partial_z U(x(s), z(s)) \cdot l(x(s), \tilde{u}) + \ln\gamma \cdot U(x(s), z(s)) \geq \theta.$$

Consider $V - U$ has a local minimum at $(x_0, z_0)$, we will have that

$$\min_{\tilde{u}\in\mathcal{U}} \big[\gamma^h V(x(t_0 + h), z(t_0 + h)) - V(x_0, z_0)\big]$$

$$\geq \min_{\tilde{u}\in\mathcal{U}} \big[\gamma^h U(x(t_0 + h), z(t_0 + h)) - U(x_0, z_0)\big]$$

$$= \min_{\tilde{u}\in\mathcal{U}} \big[(\nabla_x U(x(t_0), z(t_0)) \cdot f(x(t_0), \tilde{u}) - \partial_z U(x(t_0), z(t_0)) \cdot l(x(t_0), \tilde{u}) + \ln\gamma \cdot U(x(t_0), z(t_0)))h\big]$$

$$\geq \theta h$$

Based on this derivation, we finally have that

$$\min_{\tilde{u}\in\mathcal{U}} \gamma^h V(x(t_0 + h), z(t_0 + h)) \geq V(x_0, z_0) + \theta h > V(x_0, z_0).$$

However, we know that Lemma 3.2 implies that

$$\min_{\tilde{u}\in\mathcal{U}} \gamma^h V(x(t_0 + h), z(t_0 + h)) \leq V(x_0, z_0),$$

which is a contradiction. Thus, we prove that

$$\max\Big\{c(x_0)-U(x_0, z_0), \ \min_{u\in\mathcal{U}}\big[\nabla_x U(x_0, z_0)\cdot f(x_0, u_0)-\partial_z U(x_0, z_0)\cdot l(x_0, u_0)+\ln\gamma\cdot U(x_0, z_0)\big]\Big\} \leq 0.$$

Hence, we prove that $V(x, z)$ is the viscosity solution to the epigraph-based HJB PDE. The uniqueness follows Theorem 1 of Chapter 10 in Evans (2022). $\qquad\square$

## A.4 Advantage Function

We define the $Q(x_t, z_t, u_t) = \max\{c(x_t), r^h V(x_{t+h}, z_{t+h}\}$ over a short time interval $h > 0$ and compute

$$
\begin{aligned}
Q(x_t, z_t, u_t) - V(x_t, z_t) =& \max\{c(x_t), r^h V(x_{t+h}, z_{t+h})\} - V(x_t, z_t) \\
=& \max\{c(x_t) - V(x_t, z_t), (1 + \ln\gamma h)(V(x_t, z_t) + \nabla_{x_t} V \cdot f(x_t, u_t)h \\
& - \partial_{z_t} V \cdot l(x_t, u_t)h - V(x_t, z_t) + o(h)\} \\
=& \max\{c(x_t) - V(x_t, z_t), (\nabla_{x_t} V \cdot f(x_t, u_t) - \partial_{z_t} V \cdot l(x_t, u_t) + \ln\gamma \cdot V)h\}
\end{aligned}
$$

We divide $h$ on both sides of the above equation and let $h \to 0$ to compute the advantage function as

$$
\begin{aligned}
A(x_t, z_t, u_t) &= \lim_{h \to 0} \frac{Q(x_t, z_t, u_t) - V(x_t, z_t)}{h} \\
&= \max\{c(x_t) - V(x_t, z_t), \nabla_{x_t} V \cdot f(x_t, u_t) - \partial_{z_t} V \cdot l(x_t, u_t) + \ln\gamma \cdot V\}
\end{aligned}
$$

## A.5 Convergence of Epigraph Value Function

Consider the augmented state $(x, z)$ with state constraint $c(x)$ and non-negative cost $l(x, u)$. Define the discounted epigraph-Bellman operator over a short step $\Delta t > 0$

$$
(\mathcal{T}V)(x_t, z_t) := (1 - \gamma^{\Delta t})c(x_t) + \gamma^{\Delta t} \min_{u \in \mathcal{U}} \left\{ \max\left\{ c(x_t), V(x_{t+\Delta t}, z_{t+\Delta t}) \right\} \right\},
$$

for $V : \mathcal{X} \times \mathbb{R} \to \mathbb{R}$ bounded. Then the value iteration $V_{k+1} = \mathcal{T}V_k$ converges uniformly to the unique fixed point of $\mathcal{T}$.

*Proof.* (i) Contraction. For any $c(x_t)$ and bounded functions $V, W$, we have the following condition satisfying the contraction.

$$
\begin{aligned}
&|\max\{c(x_t), V(x_{t+\Delta t}, z_{t+\Delta t})\} - \max\{c(x_t), W(x_{t+\Delta t}, z_{t+\Delta t})\}| \\
\leq& |V(x_{t+\Delta t}, z_{t+\Delta t}) - W(x_{t+\Delta t}, z_{t+\Delta t})| \\
\leq& \|V - W\|_\infty
\end{aligned}
$$

(ii) Existence and uniqueness. By Banach's fixed-point theorem, $\mathcal{T}$ admits a unique fixed point $V$, and for value iteration $V_{k+1} = \mathcal{T}V_k$ we have that

$$
\|V_k - V\|_\infty \leq \gamma^k \|V_0 - V\|_\infty \to 0,
$$

(iii) Approximate evaluation. If each iteration uses an approximate operator $\widetilde{\mathcal{T}}$ satisfying $\|\widetilde{\mathcal{T}}V - \mathcal{T}V\|_\infty \leq \varepsilon$, then

$$
\limsup_{k \to \infty} \|V_k - V\|_\infty \leq \frac{\varepsilon}{1 - \gamma^{\Delta t}}.
$$

$\square$

# B Training Algorithms

In this part, we provide additional details on the overall algorithmic pipeline and clarify the key implementation choices.

---

**Algorithm 1** Epigraph-Based Continuous-Time MARL

---

1: Initialize actor $\pi_\theta$, return critic $V_\psi^{\text{ret}}$, constraint critic $V_\phi^{\text{cons}}$, dynamics network $f_\xi$, reward network $l_\varphi$, and local rollout $\mathcal{R}$.
2: **for** $l = 1, \ldots, T$ **do**
3:     ▷ **Collect one rollout:**
4:     $x \leftarrow \text{env.reset}()$
5:     **for** $k = 1, \ldots, K$ **do**
6:         sample arbitrary decision time $t \sim \mathcal{T}$
7:         **for** each agent $i = 1, \ldots, N$ **do**
8:             $u_i \sim \pi_{\theta_i}(u_i \mid x)$
9:         **end for**
10:         set joint action $u = (u_1, \ldots, u_N)$
11:         $(x', r) \leftarrow \text{env.step}(u)$
12:         append $(x, u, r, x')$ to local rollout $\mathcal{R}$
13:         $x \leftarrow x'$
14:     **end for**
15:     ▷ **Outer optimization: epigraph update**
16:     find $z^* = \inf\{z \in \mathbb{R} : \max\{V_\phi^{\text{cons}}(x), V_\psi^{\text{ret}}(x, z) - z\} \le 0\}$
17:     ▷ **Dynamics and Cost Model learning on** $\mathcal{R}$
18:     update $\xi, \varphi$ as per the Eq. 17.
19:     ▷ **Inner optimization given** $z^*$**: Critic update on** $\mathcal{X}_R$
20:     update $\psi, \phi$ by losses $\mathcal{L}_{\text{cons}}, \mathcal{L}_{\text{ret}}, \mathcal{L}_{\text{HJB}}$ and $\mathcal{L}_{\text{VGI}}$ as per the Eq. 11, Eq. 10 and Eq. 12.
21:     ▷ **Actor update for each agent**
22:     **for** $i = 1, \ldots, N$ **do**
23:         compute $A(x, u, z^*)$ for all $(x, u, z^*) \in \mathcal{X}_R$ and update the $\theta$ as the Eq. 18.
24:     **end for**
25: **end for**

---

## C  ENVIRONMENTAL SETTINGS

We provide detailed descriptions of all benchmark environments used in our experiments. For each scenario, we list the number of agents, the number of obstacles, the safety constraints imposed, and the specific task objective with metrics.

**Metrics.** We report two primary metrics—one reward-style *training score* that aggregates task cost and constraint penalty, and one *violation rate* measured over held-out rollouts. **(1) Cumulative penalty / reward-style training score.** In many standard environments (e.g., MPE and multi-agent MuJoCo), the task reward often consists of two independent components: (i) a *task term* such as distance-to-target or velocity tracking, and (ii) a safety penalty that is activated only when constraint-relevant events occur (e.g., collisions or proximity violations). This design is also used in prior safe MARL methods such as MACPO and Lagrangian baselines (Gu et al., 2021).

For clarity of notation, we write the task cost as $\ell_t \ge 0$ (derived from the negative reward of the task term) and denote the constraint penalty as $\kappa_t \ge 0$. The environment therefore provides a composite instantaneous cost

$$\psi_t := \ell_t + \kappa_t,$$

which simply aggregates the task objective and the constraint penalty already defined in the environment. For a trajectory $\tau$ with horizon $T(\tau)$, we define the total episode cost as

$$J(\tau) := \sum_{t=0}^{T(\tau)-1} \psi_t, \qquad S(\tau) := -J(\tau),$$

where $S(\tau)$ is the cumulative reward used for performance plots.

**(2) Violation rate (evaluation).** Given $N_{\text{eval}}$ episodes (we use $N_{\text{eval}} = 100$ by default), define the episode-level violation indicator

$$v(\tau) := \mathbf{1}\{\exists t \text{ s.t. } \kappa_t > 0\},$$

i.e., an episode is counted as violating if it ever incurs a positive state-constraint penalty.[1] The violation rate is then

$$\text{Viol. Rate} \;=\; \frac{1}{N_{\text{eval}}} \sum_{i=1}^{N_{\text{eval}}} v(\tau_i).$$

## C.1 SAFE MPE

In the MPE, we setup the details as follows: **Action.** Continuous 2-D acceleration for x and y axis. **Reward and costs.** Each agent is assigned a per-agent target $g_i$. The dense goal reward is

$$r_i^{\text{goal}}(t) = -\|x_i(t) - g_i\|_2.$$

A discrete collision cost with obstacles or other agents applies:

$$c_i^{\text{disc}}(t) = \begin{cases} 10, & \text{if agent–obstacle overlap} \\ 0, & \text{otherwise.} \end{cases}$$

We also record a continuous proximity/penetration cost (not added into the dense goal reward):

$$c_i^{\text{cont}}(t) \;=\; \frac{1}{2}\sum_{o\in\mathcal{O}}\phi\big((r_i + r_o) - \|x_i - x_o\|\big), \quad \phi(\delta) = \begin{cases} 20\,\delta, & \delta > 0 \;\;(\text{overlap}) \\ 0.5\,\delta, & \delta \le 0 \end{cases}$$

where $r_i, r_o$ are radius (sizes).

**Difference from the original discrete-time MPE.** The standard MPE environment uses a fixed and discrete integration step $\Delta t$, where each simulation step updates the agent states according to $p_{t+1} = p_t + v_t \Delta t$ and $v_{t+1} = v_t + f_t \Delta t$ with a fixed time increment. In contrast, our continuous-time MPE adapts the physical integration step to an arbitrary $\Delta t$ provided by the learning algorithm. The state evolution follows

$$\dot{p}(t) = v(t), \qquad \dot{v}(t) = \frac{f(t)}{m} - \text{damping} \cdot v(t),$$

and is numerically integrated via

$$p \leftarrow p + v \cdot \Delta t, \qquad v \leftarrow v + \frac{f}{m}\Delta t,$$

using the user-specified $\Delta t$. For clarity, the update used in the original environment is

$$\texttt{step}(F):$$
$$p = p + v \cdot 0.1 \quad \text{(fixed as 0.1),}$$
$$v = v + \frac{F}{m} \cdot 0.1 \quad \text{(fixed as 0.1).}$$

Our continuous-time version introduces

$$\texttt{step\_continuous}(F, \Delta t):$$
$$p = p + v \cdot \Delta t \quad \text{(depend on the input } \Delta t),$$
$$v = v + \frac{F}{m} \cdot \Delta t \quad \text{(depend on the input } \Delta t).$$

so that the state update depends directly on the argument $\Delta t$ rather than a fixed constant.

**Corridor.** This scenario contains 3 agents with 2 large corridor walls. Agents must avoid collisions with the corridor walls and with each other while navigating from their starting positions to reach the assigned target locations on the opposite side.

---

[1]If $\kappa_t$ is an indicator of hard violations, this coincides with "any violation." If $\kappa_t$ is a continuous hinge, we use the same criterion $\kappa_t > 0$.

**Formation.** This scenario also involves 3 agents and 2 obstacles. The agents are required to bypass obstacles and then coordinate to form a triangular formation at the designated region, under the constraint of avoiding collisions with both obstacles and other agents.

**Line.** In this task, 3 agents operate in an environment with 2 obstacles. After avoiding the obstacles, the agents must position themselves to form a straight line. The safety constraints enforce that no agent collides with obstacles or with other agents during navigation.

**Target.** This scenario uses 2 agents with 1 obstacle placed in the environment. Each agent is assigned a fixed target position, and the agents must navigate to their respective goals while avoiding collisions with the obstacle and with each other.

**Cooperative Navigation.** This is a cooperative navigation task with 3 agents and no obstacles. The agents must spread out to cover multiple target landmarks while avoiding collisions among themselves. Specifically, the agents' goals are the one closest to them rather than fixed ones.

**Cooperative Predator–Prey.** This task includes 3 controllable predator agents and 1 prey that moves randomly. There are no obstacles, but predators must avoid colliding with each other. The predators' objective is to coordinate their movements to capture the prey.

### C.2 SAFE MULTI-AGENT MuJoCo

**Half Cheetah.** We adapt the Half Cheetah environment into three multi-agent variants: Half Cheetah-2x3, Half Cheetah-3x2, and Half Cheetah-6x1. In each case, the body is partitioned into joints agents with different grouping configurations. For example, Half Cheetah-3x2 is three agents with 2 moving joints for each agent. Randomly placed walls are introduced into the environment, requiring the agents not only to coordinate efficient forward locomotion but also to avoid collisions with obstacles.

**Reward.** $r = r_{\text{run}} = \frac{x_{t+1} - x_t}{\Delta t}$.

**Safety cost.** A binary proximity cost to the wall:
$$c_t = \mathbf{1}\{\, |x_{\text{wall}} - x_{\text{agent}}| < 9 \,\} \in \{0, 1\}.$$
Observation augments the usual state with wall velocity, wall force proxy, and clipped distance to the wall; the environment also recolors the wall when unsafe.

**Difference from the original MuJoCo environment.** In standard MuJoCo control tasks, the simulation uses a fixed micro time step $0.01$ (each frame takes $0.01$), and each environment step corresponds to a fixed number of internal physics frames (e.g., `frame_skip = 5`), resulting in a fixed control interval $\Delta t = 0.05$. Our continuous-time MuJoCo variant removes this fixed control interval. For any desired $\Delta t$, we execute
$$\texttt{do\_simulation}(a, N), \qquad N = \frac{\Delta t}{0.01},$$
i.e., the number of internal physics frames is chosen dynamically according to the requested integration step. Thus the effective control interval is no longer fixed but fully determined by $\Delta t$, enabling variable-resolution continuous-time rollouts. The reward terms (forward velocity, control cost, contact cost) are normalized by the actual $\Delta t$, ensuring consistency across different temporal resolutions. The original update is
$$\texttt{step}(u): \qquad N = 5, \quad \texttt{do\_simulate}(u, N).$$
Our continuous-time version becomes
$$\texttt{step\_continuous}(u, \Delta t): \qquad N = \Delta t/0.01, \quad \texttt{do\_simulate}(u, N).$$

**Ant.** We construct four multi-agent variants of the Ant: Ant-2x4, Ant-4x2, Ant-8x1, and Ant-2x4d. In all cases, the body is controlled by joints agents arranged in different groupings across the legs. As with Half Cheetah, walls are introduced as obstacles, and the agents must coordinate locomotion while ensuring safety by avoiding collisions with these obstacles. The reward is set the same as the Half Cheetah.

**Safety shaping.** Identical piecewise-slant corridor: compute $y_{\text{off}}$ from $(x, y)$ and define
$$c_t^{\text{obj}} = \mathbf{1}\{|y_{\text{off}}| < 1.8\}.$$

C.3 CONSTRAINED COUPLED OSCILLATOR ENVIRONMENT

We consider a two–agent coupled spring–damper system. The state and control are

$$x = [x_1 \quad v_1 \quad x_2 \quad v_2]^\top, \qquad u = [u_1 \quad u_2]^\top.$$

Each agent $i \in \{1, 2\}$ controls one mass with continuous–time dynamics

$$\dot{x}_i = v_i,$$
$$\dot{v}_i = -k\,x_i - b\,v_i + u_i,$$

with spring constant $k = 1.0$ and damping coefficient $b = 0.5$. Stacking the states gives $\dot{x} = Ax + Bu$ with

$$A = \begin{bmatrix} 0 & 1 & 0 & 0 \\ -k & -b & 0 & 0 \\ 0 & 0 & 0 & 1 \\ 0 & 0 & -k & -b \end{bmatrix}, \qquad B = \begin{bmatrix} 0 & 0 \\ 1 & 0 \\ 0 & 0 \\ 0 & 1 \end{bmatrix}.$$

**Control limits and discretization.**

Actions are normalized $\tilde{u}_i \in [-1, 1]$ and mapped to physical inputs by $u_i = u_{\max}\tilde{u}_i$ with $u_{\max} = 10$ (component–wise box constraint).

$$v_i^{t+1} = v_i^t + \big( -k\,x_i^t - b\,v_i^t + u_i^t \big)\Delta t,$$
$$x_i^{t+1} = x_i^t + v_i^{t+1}\Delta t,$$

for a horizon of $N = 30$ steps.

**Stage cost.** The per–step quadratic cost is

$$\ell(x, u) = x_1^2 + x_2^2 + \lambda_c\,(x_1 - x_2)^2 + \beta\,(u_1^2 + u_2^2),$$

with coupling strength $\lambda_c = 2.0$ and control penalty $\beta = 0.01$. Equivalently, $\ell(x, u) = x^\top Q x + u^\top R u$ where

$$Q = \begin{bmatrix} 1 + \lambda_c & 0 & -\lambda_c & 0 \\ 0 & 0 & 0 & 0 \\ -\lambda_c & 0 & 1 + \lambda_c & 0 \\ 0 & 0 & 0 & 0 \end{bmatrix}, \qquad R = \beta I_2.$$

For training we use a shaped reward

$$r_t = -\frac{1}{30}\,\ell(x_t, u_t).$$

**Hard state constraint.** We impose an ordering constraint between the two positions,

$$x_1 \leq x_2 + 0.02,$$

and record an additional penalty

$$p_t = -10 \cdot \mathbf{1}\{\, x_{1,t} > x_{2,t} + 0.02 \,\},$$

returned alongside $r_t$.

**Smooth violation signal (for logging).** We also log a smooth surrogate of the constraint violation,

$$\phi(x) = 2\,\sigma\big(s\,(x_1 - x_2 + 0.02)\big) - 1, \qquad \sigma(z) = \frac{1}{1 + e^{-z}}, \quad s = 20,$$

which maps to $(-1, 1)$ and grows monotonically with the amount of violation.

**Unconstrained LQR.** The continuous-time algebraic Riccati equation (CARE)

$$A^\top P + PA - PBR^{-1}B^\top P + Q = 0$$

is solved for the unique positive semidefinite matrix $P$. The unconstrained optimal linear feedback is

$$K = R^{-1}B^\top P, \qquad u_{\mathrm{LQR}}(x) = -Kx.$$

**Hard state constraint and CBF condition.** We impose the safety constraint

$$x_1 - x_2 - 0.02 \leq 0 \quad \Longleftrightarrow \quad h(x) := 0.02 - (x_1 - x_2) \geq 0.$$

Let $\nabla h(x) = \begin{bmatrix} -1 & 0 & 1 & 0 \end{bmatrix}^\top$. A (first-order) control barrier function (CBF) condition enforces forward invariance of the safe set $\mathcal{C} = \{x : h(x) \geq 0\}$ by requiring

$$\dot{h}(x, u) = \nabla h(x)^\top (Ax + Bu) \geq -\alpha h(x),$$

with a user-chosen class-$\mathcal{K}$ parameter $\alpha > 0$. Defining

$$a(x) := \nabla h(x)^\top B \in \mathbb{R}^2, \qquad b(x) := -\nabla h(x)^\top Ax - \alpha h(x) \in \mathbb{R},$$

the CBF condition Eq. C.3 is the single affine-in-$u$ half-space constraint

$$a(x)^\top u \geq b(x).$$

**Closed-form safety projection.** To obtain a safe control with minimal distortion from $u_{\text{LQR}}$, we solve the weighted projection

$$\min_{u \in \mathbb{R}^2} \tfrac{1}{2} (u - u_{\text{LQR}})^\top W (u - u_{\text{LQR}}) \quad \text{s.t.} \quad a(x)^\top u \geq b(x),$$

with $W = R$ ("$R$-metric"; Euclidean $W = I$ is also possible). Because Eq. C.3 has a single linear constraint, it admits a closed form:

$$u^\star(x) = \begin{cases} u_{\text{LQR}}(x), & \text{if } a^\top u_{\text{LQR}} \geq b, \\ u_{\text{LQR}}(x) + \tau W^{-1} a, & \text{otherwise, with } \tau = \dfrac{b - a^\top u_{\text{LQR}}}{a^\top W^{-1} a}. \end{cases}$$

Finally we saturate to the actuator limits $u_{\max} > 0$:

$$u_{\text{GT}}(x) = \text{clip}(u^\star(x), -u_{\max}, u_{\max}).$$

# D  ADDITIONAL ENVIRONMENTAL RESULTS

## D.1  VISUAL TRAJECTORIES

The trajectory demonstrations in Fig. 10 highlight clear behavioral differences across algorithms in Formation scenario. Our proposed method EPI learns smooth trajectories that avoid obstacles while consistently reaching the target, demonstrating both constraint satisfaction and goal achievement. In contrast, EPPO occasionally captures the avoidance behavior but often gets stuck at suboptimal solutions. This is because during training, its randomized sampling of the auxiliary state $z$ prevents stable policy convergence in continuous-time settings; even if outer optimization is applied at execution, the learned policy lacks accurate control signals. On the other hand, MACPO, which enforces hard constraints via a trust-region style update, tends to overestimate the obstacle region. As a result, agents often exhibit overly conservative behaviors—such as retreating toward corners to avoid violations—rather than efficiently pursuing their targets. Together, these comparisons confirm that EPI achieves the most balanced and effective behavior among the three approaches.

## D.2  PERFORMANCE UNDER STOCHASTIC SETTINGS

To evaluate robustness under stochastic dynamics, we perturb the continuous-time transition model as $x_{t+\Delta t} = f(x_t, u_t) \Delta t + \varepsilon_t, \varepsilon_t \sim \mathcal{N}(0, \sigma^2 I)$. in Fig.11. We consider three noise magnitudes: **Low Noise:** $\sigma^2 = 0.1$ **Mid Noise:** $\sigma^2 = 0.5$ and **High Noise:** $\sigma^2 = 1.0$. We observe that No Noise and Low Noise yield similar identical cost and constraint-violation behavior across all three tasks. Because the PINN-based value approximation are inherently robust to small local perturbations, as long as the injected disturbance is within a moderate range, the learned dynamics model, cost model, and value gradients remain accurate. In contrast, Mid Noise and High Noise introduce much larger deviations in the state propagation. These disturbances accumulate over time, causing the PINN to receive significantly deviated training signals. Since our method does not incorporate explicit uncertainty modeling or stochastic HJB formulations, the serious noise directly degrades the learned critic and value gradients, eventually leading to unstable or even failed policies.

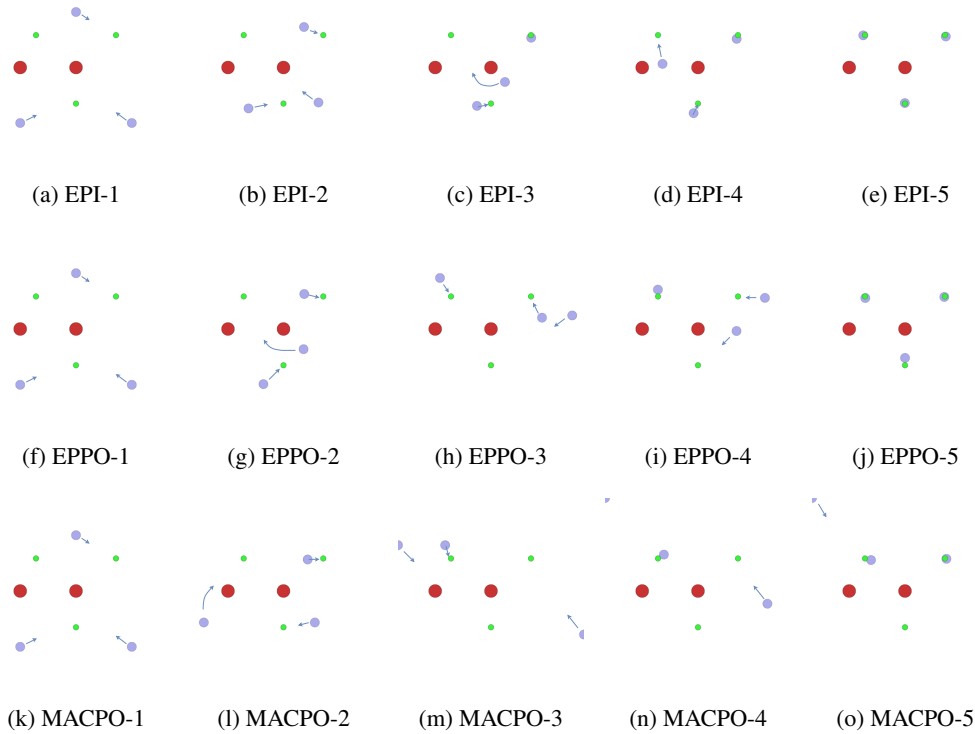

Figure 10: Trajectory demonstrations (key frames) across methods in Formation. Row 1: EPI results, Row 2: EPPO results, Row 3: MACPO results.

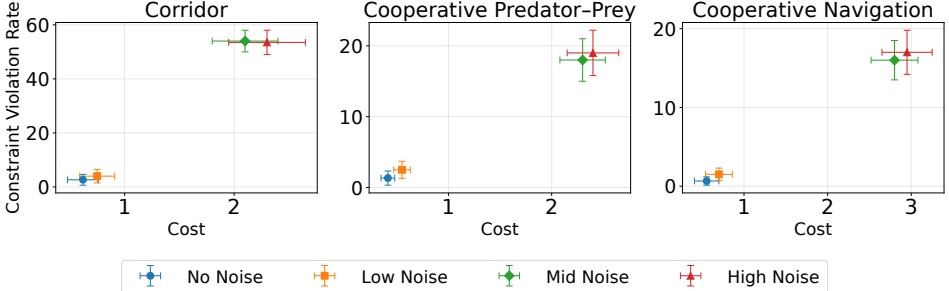

Figure 11: Performance under different noise levels.

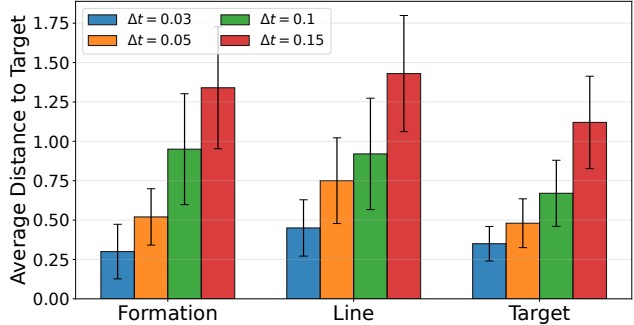

Figure 12: Average distance to the target under different $\Delta t$.

## D.3 EFFECT OF THE DISCRETIZATION INTERVAL.

Figure 12 evaluates how the choice of discretization interval $\Delta t$ affects the performance of EPI. For each fixed $\Delta t$, we roll out complete trajectories using the learned policy and measure the average distance to the target over the entire trajectory. Across all three scenarios, we observe a consistent trend: the *average distance to the target increases as $\Delta t$ becomes larger*. This behavior is expected in continuous-time control. When $\Delta t$ is small, the temporal resolution is high and the policy is updated frequently, allowing the learned value gradients to provide fine-grained control corrections. In contrast, larger $\Delta t$ leads to *coarser control updates*, reducing the precision of the policy's response to the evolving system dynamics. Moreover, both the HJB residual and the VGI update rely on local differential information. As $\Delta t$ grows, the mismatch between the continuous-time formulation and the discrete rollout increases, which in turn amplifies approximation errors in the learned value gradients. These errors accumulate along the trajectory and result in the observed degradation in task accuracy.

## D.4 TRAJECTORY OF $z^*$ THROUGH THE TRAINING

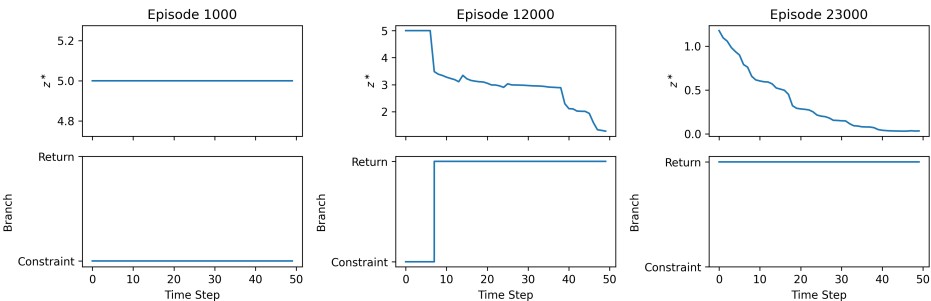

Figure 13: $z^*$ trajectory through the training in the target.

Figure 13 illustrates the evolution of the optimal epigraph variable $z_t^*$ and the active branch (`return` vs. `constraint`) at three representative stages of training. In early training (Episode 1000), the policy frequently visits infeasible states, causing $V_{\text{cons}}(x_t) > 0$ and forcing the epigraph to select the constraint branch; consequently $z_t^*$ remains at the clipped upper bound $z_{\text{max}}$. By mid training (Episode 12000), the critic starts to maintain $V_{\text{cons}}(x_t) \leq 0$ for part of the trajectory, producing intermittent switching and a decreasing $z_t^*$. In late training (Episode 23000), the trajectory remains feasible, the return branch is consistently selected, and $z_t^*$ decreases smoothly along the rollout. These behaviors align with the expected epigraph semantics: infeasible states produce $z_{\text{max}}$, while improved policies yield stable return-dominated gradually decreasing $z_t^*$.

## D.5 COMPARE EPI WITH TRADITIONAL EPIGRAPH METHOD

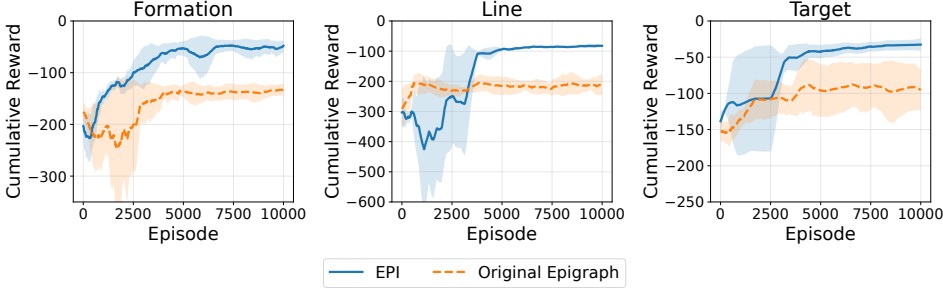

Figure 14: Performance of EPI and traditional epigraph under MPE settings.

Figure 14 compares our $z$-independent epigraph formulation (EPI) with the traditional $z$-dependent epigraph used in EPPO-like methods on the FORMATION, LINE, and TARGET tasks. In the traditional design, a scalar $z$ is randomly sampled at the initial state of each episode and then propagated

through its auxiliary dynamics, so that both critic and actor are conditioned on this randomly chosen epigraph level. As shown in Fig. 14, converges to a lower cumulative reward, and exhibits substantially larger variance across seeds. In contrast, EPI learns $z$-independent critics $\left(V^{\mathrm{cons}}(x), V^{\mathrm{ret}}(x)\right)$ and computes $z^*$ via a one-dimensional search during training, while the actor depends only on the physical state $x$. This removes the nonstationary noise introduced by random $z$ sampling: for a fixed $x$, the policy gradient under EPI is unique, whereas in the traditional epigraph it fluctuates with the sampled $z$ even when the critic has already converged. In continuous-time settings this issue is amplified, since small changes in $z$ shift the switching time between the constraint and return branches and thereby alter the entire rollout.

## D.6 Comparison between EPI and Discrete-time Baselines

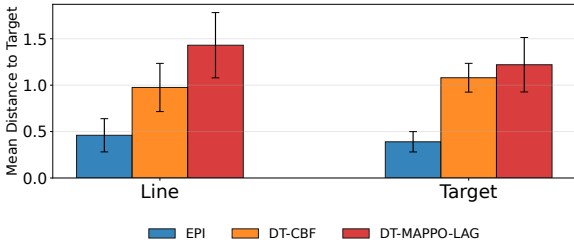

Figure 15: Performance of EPI and discrete-time baselines under MPE settings.

To validate the performance of traditional discrete-time based methods in continuous-time settings, the Fig 15 compares EPI with two discrete-time baselines (DT-CBF and DT-MAPPO-LAG) on the *Line* and *Target* tasks in the continuous-time MPE environment. All baselines are adapted to the discrete-time setting by removing their residual-loss components. Apart from this modification, all implementation details follow their original published versions (Zhang et al., 2025a). Across both tasks, EPI consistently achieves lower mean distance to the target and smaller variance, demonstrating the performance gain from the modules that designed for the continuous-time settings.

Table 1: Hyperparameter settings used.

| Parameter | Value |
|---|---|
| Episode length for MPE | 50 |
| Episode length for MuJoCo | 100 |
| Episode length for Didactic | 50 |
| Total number of episode for MPE | 30000 |
| Total number of episode for MuJoCo | 30000 |
| Total number of episode for Didactic | 3000 |
| z range for MPE | 0-10 |
| z range for MuJoCo | 0-5 |
| z range for Didactic | 0-2 |
| Discount factor $\gamma$ | 0.99 |
| Actor learning rate | 0.0001 |
| Critic (Return) learning rate | 0.001 |
| Critic (Constraint)learning rate | 0.001 |
| Dynamics model learning rate | 0.001 |
| Reward model learning rate | 0.001 |
| Exploration steps | 1000 |
| Model save interval | 1000 |
| Random seed | 113-120 |

## E  HYPERPARAMETERS AND NEURAL NETWORK STRUCTURES

Experiments were conducted on hardware comprising an Intel(R) Xeon(R) Gold 6254 CPU @ 3.10GHz, four NVIDIA A5000 GPUs and eight NVIDIA A6000 GPUs. This setup ensures the computational efficiency and precision required for the demanding simulations involved in multi-agent reinforcement learning and safety evaluations.

Table 1 lists the defaults used in all experiments. Episode lengths are chosen so that a single rollout covers a full interaction cycle (50 steps for MPE and the didactic environment, 100 for MuJoCo). We train for 30000 episodes in MPE and MuJoCo and for 3000 episodes in the didactic setting, reflecting simulator cost and convergence speed. The z range controls epigraph sampling for the VGI updates and is set wider in MPE (0–10), moderate in MuJoCo (0–5), and narrow in the didactic task (0–2). The actor uses a conservative learning rate (1e-4) for stable policy updates; the critics and the dynamics/reward models use 1e-3 to accelerate value/model fitting. Training is warm-started with 1000 exploration steps, checkpoints are saved every 1000 episodes, and reported results are averaged over seeds 113–120.

Table 2: Summary of neural network architectures used in our framework.

| Network | Input Dimension | Architecture and Activation |
| --- | --- | --- |
| Return Value Network | State ($d$) | FC(128) $\rightarrow$ FC(128) $\rightarrow$ FC(1), ReLU or Tanh |
| Constraint Value Network | State ($d$) | FC(128) $\rightarrow$ FC(128) $\rightarrow$ FC(1), ReLU or Tanh |
| Dynamics Network | State + Joint Action ($d + na$) | FC(128) $\rightarrow$ FC(128) $\rightarrow$ FC($d$), ReLU |
| Reward Network | State + Joint Action ($d + na$) | FC(128) $\rightarrow$ FC(128) $\rightarrow$ FC(1), ReLU |
| PolicyNet | Observation + Time Interval ($o + 1$) | FC(128) $\rightarrow$ FC(128) $\rightarrow$ FC(64) $\rightarrow$ FC($a$), ReLU |

Table 2 summarizes the five multilayer perceptrons used in our framework. Two scalar critics map the state $x \in \mathbb{R}^d$ to the return value and the constraint value, each with two hidden layers of width 128 and ReLU or Tanh activations. The dynamics and reward models take the concatenated state–action input $(x, u) \in \mathbb{R}^{d+na}$ and output, respectively, a $d$-dimensional state derivative/increment and a scalar reward; both use two 128-width hidden layers with ReLU. The policy network consumes the observation $o \in \mathbb{R}^o$ augmented with a scalar time-interval feature $\Delta t$ to condition actions on continuous-time step size, and produces an $a$-dimensional action through a 128–128–64 hidden stack with ReLU.

Notation: $d$ = state dimension, $o$ = observation dimension, $a$ = per-agent action dimension, $n$ = number of agents, so the joint action has dimension $na$. The value heads output scalars; the dynamics head outputs $\mathbb{R}^d$; the policy head outputs $\mathbb{R}^a$. Action squashing or clipping to environment bounds (if used) is applied after the final linear layer.

## F  THE USE OF LARGE LANGUAGE MODELS (LLMS)

We employed LLMs as a writing assistant to polish the paper.

