# OpenReview forum: "Safe Continuous-time Multi-Agent Reinforcement Learning via Epigraph Form"
_ICLR.cc/2026/Conference — ICLR 2026 Poster_

### Official Review · Reviewer_cjAT · 2025-10-29

**Soundness:** 3
**Presentation:** 1
**Contribution:** 2
**Rating:** 4
**Confidence:** 4

**Summary:**

This paper considers the continuous-time multi-agent reinforcement learning (CT-MARL) problem with safety constraints. To address the problem, the paper presents a framework that transforms discrete MDPs into CT-CMDPs via an epigraph-based reformulation. The problem is then solved with a PINN-based actor-critic method. Empirical results show that the proposed method performs better in continuous-time safe multi-particle environments and safe multi-agent MuJoCo environments.

**Strengths:**

1. The problem considered is interesting.

2. The proposed method shows strong empirical results.

3. Each part of the loss function is well ablated.

**Weaknesses:**

1. The proposed method is not very well motivated and includes some unclear parts. Please refer to Questions.

1. The metric used in the experiment section can be questionable. The paper "designs the reward as the summation of the minus cost and constraints", which can be sensitive to the relative scales of the cost and the constraints. I think figures like Figure 4 are clearer than Figure 3. Or the authors can also show the training curves of the cost and the constraints separately.

1. It seems that all the baselines chosen in this paper are designed for discrete-time MDPs. It would be better if some continuous-time methods could be included as baselines.

1. Minor: Some acronyms are not defined clearly in the paper, e.g., PINN, PDE.

**Questions:**

1. How are the decision times $t_k$ selected? Are they given, or does the policy need to decide?

1. Considering the motivation of the proposed method, why does the value become discontinuous when state constraints are violated? Why can this hinder the convergence? Why can the epigraph formulation address this issue?

1. Why does the random sampling of $z$ introduce nonstationary noise and lead to poor convergence? Is this a drawback only in the continuous-time setting? It would be great if the authors could show this theoretically or empirically, or both.

1. It seems that the value function $\tilde V$ and the advantage function $A$ are conditioned on $z^\*$, which, to my understanding, can change a lot during training. However, the learned policies are not conditioned on $z^*$. Therefore, it seems that the policy learning target changes constantly during training. Will this cause unstable training?

1. Why $f_\xi$ and $l_\phi$ do not have $\Delta_t$ as input in Equation (16)?

1. Which part of the proposed method is designed specifically for *multi-agent* systems?

---

> ### Author Response · Authors · 2025-11-24
> **We appreciate the reviewer's time and detailed comments. We provide  point-by-point responses below: Reply 1**
>
> **W1. The metric used in the experiment section can be questionable.**
>
> We want to clarify we do not introduce a new reward definition, nor do we modify the environment's native metrics. The reward used in our experiments is exactly the one provided by each environment (e.g., distance to the target plus collision penalties in MPE), which is the same structure adopted in MACPO and other
> constrained-RL baselines.
>
> Our intention of the original wording was to highlight that, for analysis purposes of the epigraph formulation, we separate the environment signal into (i) the state constraints component (e.g., the collision penalty) and (ii) the task-related cost (e.g., the distance to the target). The sentence “summation of the minus cost and constraints” was therefore misleading, and we have revised the manuscript accordingly in section 4.2 Q1 (highlighted in blue).
>
> Besides, the results that constraint and cost terms are disentangled are also provided in Figure 3 and 5.
>
> **W2. It seems that all the baselines chosen in this paper are designed for discrete-time MDPs, It would be better if some continuous-time methods could be included as baselines.**
>
> We thank the reviewer for raising this point. To the best of our knowledge,
> there is currently no existing work on safe multi-agent reinforcement learning formulated directly in continuous time. And for current baselines, we already adapted to the continuous settings by adding up the loss terms like residual loss.
>
> In addition, we further implemented a continuous-time CBF (CT-CBF) baseline inspired by the robust CBF framework of [1]. The method in [1] enforces a one-step discrete-time barrier condition
> $
> \nabla B_\theta(x)^\top f_\xi(x,u) + \alpha B_\theta(x) \le 0,
> $
> where each agent learns a dynamics model $f_\xi$, a barrier network $B_\theta$, and a policy network. This yields a faithful CT adaptation of the original CBF framework.
>
> CBF shows more conservative behavior than EPI. A key reason is that continuous-time CBF conditions require accurate approximations of $\nabla B(x)$; in multi-agent systems with high-dimensional coupled states, these gradients are difficult to approximate reliably. Small approximation errors in $\dot{B}(x) = \nabla_x B(x)\cdot f(x,u)$ can drastically distort the effective safe set.
>
> [1] Emam, Y., Notomista, G., Glotfelter, P., Kira, Z., & Egerstedt, M. (2022). Safe reinforcement learning using robust control barrier functions. IEEE Robotics and Automation Letters
>
> **W3. Some acronyms are not defined clearly in the paper, e.g., PINN, PDE.**
>
> We thank the reviewer for pointing this out. We have revised the manuscript accordingly.
>
> **Q1. How are the decision times $t_k$ selected? Are they given, or does the policy need to decide?**
>
> The decision times $\{t_k\}$ are given. In our continuous-time formulation, the integration step $\Delta t_k = t_{k+1}-t_k$ varies across the rollout, but the overall episode horizon is kept fixed. Although the policy does not choose the decision times themselves, it must condition its action on the current $\Delta t_k$. Intuitively, different $\Delta t_k$ correspond to different effective control durations, for example, a larger $\Delta t_k$ requires more cautious actions to avoid overshooting targets or violating constraints.
>
> **Q2. Considering the motivation of the proposed method, why does the value become discontinuous when state constraints are violated? Why can this hinder the convergence? Why can the epigraph formulation address this issue?**
>
> When state constraints are violated, the true value function becomes non-smooth. This arises because constraint violations typically incur an instantaneous jump in the value function (e.g., collision penalties inject a large negative value). Such non-smooth structure is hard for neural networks including PINNs to approximate. In practice, they struggle to fit the abrupt changes in $V(x)$ near the constraint boundary, which leads to unstable value learning and then degrades the performance.
>
> The epigraph formulation is a technique that smooth the value function, it alleviates this difficulty by smoothing the constraints first then use max operator to combine both the constraints and cost (which is also a smooth operation).
>
> Consequently, the epigraph formulation yields a value structure that is smooth and substantially easier for neural networks to approximate, while still preserving the correct CMDP semantics.

---

> ### Author Response · Authors · 2025-11-25
> **Reply 2**
>
> **Q3. Why does the random sampling of $z$ introduce nonstationary noise and lead to poor convergence?**
>
> We thank the reviewer for the insightful question. In the previous epigraph-based methods, a scalar $z$ is sampled at the initial state and then propagated forward according to its own dynamics. The baseline EPPO follows the same paradigm where both the critic and the actor take $z$ as input and therefore become $z$-dependent.
>
> However, this causes a fundamental stability issue. Because the epigraph value $V(x,z)=\max ( V_{\mathrm{ret}}(x)-z, V_{\mathrm{cons}}(x) )$ has two branches, different sampled values of $z$ activate different branches, leading to different policy gradients. Small $z$ typically activates the return branch, while a slightly larger $z$ activates the constraint branch. When $z$ is randomly sampled, these branch switches occur frequently and introduce strong non-stationarity into the actor update. This problem becomes more severe in continuous time: policy optimization must align with $\partial_z V$, meaning that even small fluctuations in $z$ cause large changes in value gradients over time. We believe this is why continuous-time $z$-dependent epigraph methods become unstable. Notably, this instability has also been independently observed in another recent continuous-time epigraph paper [2].
>
> To verify this effect, we added an experiment comparing the $z$-sampled formulation against our $z$-independent epigraph formulation in Figure 14 (Please refer the Appendix D.5). Our method achieves higher performance and much lower variance across seeds, confirming that eliminating the $z$-dependent actor helps stable policy learning in continuous time.
>
> [2] Tayal, M., Singh, A., Kolathaya, S., & Bansal, S. (2025). MAD-PINN: A Decentralized Physics-Informed Machine Learning Framework for Safe and Optimal Multi-Agent Control. arXiv:2509.23960.
>
> **Q4. It seems that the value function $\tilde{V}$ and the advantage function $A$ are conditioned on $z^{*}$, which, to my understanding, can change a lot during training.**
>
> The auxiliary value $\tilde V$ and advantage $A$ are indeed conditioned on the $z^{\ast}$, but (1) the critic consists of two networks $V_{\text{ret}}(x)$ and $V_{\text{cons}}(x)$, neither of which takes $z$ or $z^{\ast}$ as input. The critic updates are therefore $z$-independent. Also, the $z^{\ast}$ is not changing randomly but  determined by $V_{\text{ret}}(x)$ and $V_{\text{cons}}(x)$. Importantly, in the standard actor-critic paradigm, the critic is also changing during the training to reflect the quality of the current policy. Similarly, our critic networks reflect the (cost + constraints), and $z*$ is totally determined by the critic networks. Thus, the resulting changes in $\tilde{V}$ do not create extra instability beyond the usual actor–critic paradigm.
>
> (2) Since the advantage uses $z^{\ast}$, its gradients are stable. The active branch is selected via
> $
> \chi(x) = \mathbf{1}(V_{\text{ret}}(x)-z^{*} \ge V_{\text{cons}}(x)),
> $
> and the derivative $\partial_z \tilde V$ appearing inside the policy gradient is
> a simple constant (either $0$ or $-1$ after the chain rule). This ensures that
> the policy never encounters highly sensitive or unstable gradients.
>
> (3) Although $z^{\ast}$ evolves during training, it does not cause oscillatory switching. This is because $z^{\ast}$ is determined by the critic values $V_{\mathrm{ret}}$ and $V_{\mathrm{cons}}$ and does not impact the switching, while in the previous epigraph-based formulation, sampled $z$ can cause branch switching directly.
>
> (4) As discussed in Q3, Using $z^{\ast}$ improves stability compared to sampled $z$.
> Previous methods sample $z$ randomly at rollout start, which frequently changes the
> active branch and injects large gradient noise. Our deterministic $z^{\ast}$ alleviates this noise source and yields more stable optimization, as shown in Fig.~14 (Appendix D.5).
>
> **Q5. Why $f_{\xi}$ and $l_{\phi}$ do not have $\Delta_{t}$ as input in Equation~(16)?**
>
> Thanks for pointing out this typo, the $\Delta t$ should be included. We updated the manuscript accordingly.
>
> **Q6. Which part of the proposed method is designed specifically for multi-agent systems?**
>
> Our method is developed for safe MARL and follows a similar setup used by mainstream safe MARL works (e.g., MACPO, MAPPO-Lag, EPPO). The multi-agent aspect lies in the formulation through the state constraints, which explicitly includes the interactions between agents such as collision avoidance. In all our experiments, the state constraint includes a collision penalty between agents. And the results show agents can avoid this collision explicitly.

---

### Official Review · Reviewer_qGRY · 2025-10-29

**Soundness:** 3
**Presentation:** 3
**Contribution:** 3
**Rating:** 4
**Confidence:** 4

**Summary:**

This paper proposes EPI, a method for solving safe continuous time MARL problems posed as a constrained optimal control problem. An epigraph reformulation is used to handle the state constraints, then a PINN is used to learn the continuous-time value function, where two additional losses are used to improve the PINN training. Empirical results on the MPE and Safe MA-MuJoCo environments demonstrate that the proposed EPI outperforms continuous-time variants of existing safe MARL methods.

**Strengths:**

- To the best of my knowledge, the use of the epigraph form for continuous-time safe MARL is novel.
- The experimental results on MPE show that EPI has strong performance compared to existing methods. On MuJoCo EPI, it is not yet clear how the performance in terms of cost and constraint compare to existing methods.

**Weaknesses:**

- Many details in the experiments section are not very clear (see questions)
- It is not clear which parts of the proposed method are claimed as novel contributions and which ones are taken from existing methods. For example, are the concepts of the target loss and value gradients claimed to be novel, or were they proposed in previous works and then adapted to the current setting?
- The experiments on multi-agent MuJoCo seem to only compare the reward (Figure 2). It is not clear how the different methods compare in terms of constraint satisfaction. Something similar to Figure 4 for the multi-agent MuJoCo setting would be appreciated.

**Questions:**

- Line 139: The paper writes that each agent applies a decentralized policy, but each policy seems to take the entire state space as input. Later on in Line 211, the authors write that the decentralized actors “map local observations”, which is confusing.
- Why do the networks take ∆t as input? Does ∆t remain constant or does it change?
- What are the weights for the different terms for the PINN learning?
- How sensitive is the method to the weighting of these terms?
- What does “overweighting” mean for the ablation in Figure 9? How much are the weights increased by?
- Algorithm 1, line 16: I’m assuming that z^* is solved for for every x in the rollout?
- How are the benchmarks “adapted”? The MPE and Safe MA-MuJoCo benchmarks are originally for discrete time.
- How does the proposed method compare to (discrete) CBF-based methods for safety in MARL, such as [A]? Though it is a discrete-time method, it should still apply to continuous-time environments, similar to how the other discrete-time methods have been adapted.

[A]: Zhang, Songyuan, et al. "Discrete GCBF Proximal Policy Optimization for Multi-agent Safe Optimal Control." ICLR 2025.

---

> ### Author Response · Authors · 2025-11-24
> **We thank the reviewer for the insightful suggestions, the detailed responses are listed below: Reply 1**
>
> We sincerely thank the reviewer for the careful reading of our papers and the insightful suggestions. We will provide detailed response below:
>
> **W1. It is not clear which parts of the proposed method are claimed as novel contributions.**
>
> Our main novelty lies in we establish the continuous-time epigraph framework for CT-CMDPs, including a new epigraph-based HJB characterization, the continuous-time outer optimization, and the resulting critic/actor learning principles. Regarding the methodological modules, not every component is newly invented. For example, value gradient iteration (VGI) originates from other work. However, our use of VGI under the epigraph-based HJB operator is new, as it requires a different switching structure and a different gradient recursion than the standard unconstrained setting. Overall, our framework itself is novel but the integrated modules are adapted rather than invented.
>
> **W2. The experiments on multi-agent MuJoCo seem to only compare the reward.**
>
> We thank the reviewer for the helpful suggestion. In the revised version, we have added the results in Figure 3. EPI consistently attains the lowest violation rates among all baselines. In contrast, existing baselines exhibit higher violations and larger variance. These results demonstrate that the proposed EPI approach effectively improves the performance in multi-agent MuJoCo tasks.
>
>
> **Q1. Line 139: the authors write that the decentralized actors “map local observations”, which is confusing.**
>
> We would like to clarify that we adopt a CTDE structure: each agent’s actor $\pi_i(o_i, \Delta t)$ takes its local observation $o_i$ as input, while the training signal (advantage value) is derived from the full state $x$. To prevent this ambiguity, we added a clarification at the end of Section 3.2.3 (highlight blue).
>
> **Q2. Why do the networks take $\Delta t$ as input? Does $\Delta t$ remain constant or does it change?**
>
> In continuous-time environments, the time interval $\Delta t$ is not fixed but is an inherent property of the system dynamics: it varies across rollouts and steps.
> Therefore, both the policy and the dynamics/reward models are condition on $\Delta t$ to produce correct predictions under different $\Delta t$ resolutions. From the policy perspective, $\Delta t$ affects the physical evolution of the system. For example, in MPE, an agent moving toward a target must account for how far it will travel within the next $\Delta t$. A larger $\Delta t$ implies a longer movement, so the optimal action may reduce speed to avoid overshooting. From the dynamics-model perspective, the next state depends on $\Delta t$ through the integration rule:
> $
> x_{t+\Delta t} = x_t + f(x_t,u_t) \Delta t.
> $
> Different $\Delta t$ values produce different next states even under the same current state and action. If $\Delta t$ is not provided as input, the dynamics network would be forced to map multiple distinct transitions to the same $(x,u)$, causing large prediction uncertainty and biased value estimation.
>
> **Q3. What are the weights for the different terms for the PINN learning?**
>
> These three losses are optimized jointly. Their weights are chosen so that the losses remain on a similar scale (typically around Residual loss = $0.4$, Target loss = $0.4$, VGI loss = $0.2$), and we select these weights using the grid search. Besides, the impact of different weight configurations is empirically evaluated in our ablation study (section 4.2 Figure 5 and 7). We clarify this joint optimization and include the weight selection strategy in the revised manuscript.
>
> **Q4. How sensitive is the method to the weighting of these terms?**
>
> The sensitivity of the weighting is listed in section 4.2, Figures 5 and 7. Specifically, the method is most sensitive to the target loss and the VGI loss.
> Removing either one causes a significant performance drop. The target loss anchors the value function in the absence of boundary conditions, preventing drift, while the VGI loss enforces accurate and stable value gradients that directly influence both critic accuracy and policy updates. While the residual loss is not that sensitive, once VGI provides reliable gradient information, the residual loss mainly acts as a structural regularizer; removing it yields only minor degradation.
>
> **Q5. What does “overweighting” mean for the ablation in Figure 9? How much are the weights increased by?**
>
> We thank the reviewer for pointing out the missing detail. "Overweighting" means multiplying the corresponding loss component by a factor of 20 while keeping the other two terms unchanged. We have relocated the original Figure 9 from Appendix to Page 10 (now Figure 7), and the corresponding detail has been updated accordingly (marked in blue).

---

> ### Author Response · Authors · 2025-11-24
> **Reply 2**
>
> **Q6. Algorithm 1, line 16: I’m assuming that $z^*$ is solved for for every x in the rollout?**
>
> Yes, for the training, our algorithm indeed solves $z^{\ast}$ for every step. Unlike prior epigraph-based methods that use $z$-dependent networks and solve for $z^*$ during execution via root-finding algorithms (which is computationally expensive), our modules are $z$-independent, which reduces the outer optimization to a one-dimensional scalar search. Therefore, even needed to compute each step at training, the cost is negligible compared with root-finding.
>
> **Q7. How are the benchmarks “adapted”?**
>
> We appreciate the reviewer’s question and have revised Appendix C.1 and C.2 (highlighted in blue) to clearly explain how the benchmarks are adapted to the continuous-time setting.
>
> Specifically, the standard MPE uses a fixed discrete step size $\Delta t$
> (e.g., $0.1$), updating states via $p_{t+1}=p_t+v_t\Delta t$ and $v_{t+1}=v_t+f_t\Delta t$. Our continuous-time MPE instead exposes $\Delta t$ as a variable parameter: $\dot{p}(t)=v(t), \dot{v}(t)=\tfrac{f(t)}{m}-\text{damping}*v(t),$ and integrates
> $
> p \leftarrow p + v\Delta t,
> v \leftarrow v + \tfrac{f}{m}\Delta t,
> $
> for an arbitrary $\Delta t$ requested during training. This enables variable-$\Delta t$ consistent with the continuous-time HJB setting.
>
> In the standard MuJoCo tasks, each environment step corresponds to a fixed number of internal physics frames (e.g., frame_skip=5), where a fixed frame interval set as $0.01$. Our continuous-time variant does not execute a fixed number of internal physics frames per environment step. Instead, for any user-specified control interval $\Delta t$, we perform
> $
> N = \frac{\Delta t}{0.01}
> $.
> For any prescribed
> $\Delta t$, we execute do_simulation(a, N), so the effective control interval is entirely determined by the requested
> $\Delta t$.
>
> **Q8. How does the proposed method compare to (discrete) CBF-based methods for safety in MARL, such as [A]?**
>
> In the revised experiments we implement a continuous-time multi-agent CBF
> variant inspired by [A] and report its performance in Figures 4 and 6. The original method enforces a one--step control barrier condition
> $
> B(x_k) \le 0  \Rightarrow
> B(f(x_k,u)) - B(x_k) + \alpha(B(x_k)) \le 0.
> $
> To adapt them in continous-time setting, we construct a CT-CBF baseline that matches our continuous-time MARL setting:
> $
> \nabla B_\theta(x)^\top f_\xi(x,u) + \alpha B_\theta(x) \le 0.
> $
> For the technical details, each agent learns a set of models tailored to the continuous-time formulation: (i) a dynamics network $f_\xi(x,u)$ trained to predict
> $(x_{t+\Delta t}-x_t)/\Delta t$, (ii) a reward (cost) network $c_\phi(x,u)$, (iii) a state-value network $V_\psi(x)$ trained with Monte--Carlo returns, and (iv) a barrier network $B_\theta(x)$. Specifically, (a) a residual loss enforcing the CBF inequality $r(x,u)=\nabla B_\theta(x)^\top f_\xi(x,u)+\alpha B_\theta(x)$. Besides, we also conduct gradient projection which is the continuous-time analogue of the one-step CBF projection used in [A], ensuring that each policy update respects the CT-CBF inequality.
>
> Empirically, CBF achieves reasonable constraint-violation levels but tends to
> be overly conservative. We hypothesize this behavior is largely due to the way CT-CBF is instantiated in our CT setting. The CBF condition relies on the gradient of a learned barrier function $\nabla B(x)$; approximation errors in this component can distort the effective safe set and degrade the performance. We have some value gradient correctness technique like VGI can help to mitigate this kind of error, but it cannot be directly transplant to CBF. But still, designing stronger CT-CBF algorithms with more accurate barrier estimation is an interesting and promising research direction.

---

### Official Review · Reviewer_GrMo · 2025-10-31

**Soundness:** 2
**Presentation:** 2
**Contribution:** 2
**Rating:** 4
**Confidence:** 3

**Summary:**

This paper proposes a novel framework for safe continuous-time multi-agent reinforcement learning (CT-MARL) using epigraph forms. The authors address the challenge of incorporating safety constraints into CT-MARL, which often leads to discontinuities in value functions that are difficult to handle with traditional methods. They introduce an epigraph-based reformulation that converts discontinuous value functions into continuous ones by augmenting the system with an auxiliary state variable z. This reformulation enables the use of Physics-Informed Neural Networks (PINNs) for stable and efficient optimization in continuous time. The authors prove the existence and uniqueness of viscosity solutions for the epigraph-based HJB PDEs, providing theoretical support for their method. Extensive experiments on continuous-time safe multi-particle environments (MPE) and multi-agent MuJoCo benchmarks demonstrate that their approach outperforms existing safe MARL baselines in terms of both cost reduction and constraint satisfaction. This work offers a robust and effective solution for safe CT-MARL by leveraging epigraph forms and PINNs to handle discontinuities and ensure stable learning in continuous time.

**Strengths:**

1. The motivation behind this work is interesting and necessary.
2. The theoretical proofs provided in this work are comprehensive.

**Weaknesses:**

1. In the section “Related work”, what are the differences between this work and [1]? I feel that this work merely extends the ideas of [1] to the continuous-time scenario. I suggest that the authors restate the contributions of this paper.
2. In the section “Methodology”: ① The figure in Fig1 is incorrect. According to the description of Algorithm 1, at least three steps are not independent but form an integrated loop. An arrow from Learning to rollout should be added. ② I don't quite understand how equation (7) is transformed into equations (8) and (9). Some additional explanations may be needed here. ③ In the “INNER OPTIMIZATION WITH CRITIC LEARNING” section, are the three loss functions updated independently or jointly? If they are updated jointly, how are their weights defined?
3. In the section “Experiments”: ① In 4.1 Benchmarks and baselines, the authors mention “In MuJoCo, we adapt several scenarios such as HalfCheetah and Ant into continuous-time versions and introduce randomly placed walls as obstacles.” I don't know how the continuous time is designed here. I think it is not necessary to spend too much space describing this environment. Instead, the differences between the original version and the continuous-time version should be explained. ② “We design the reward as the summation of the minus cost and constraints listed in Appendix C, which directly reflects performance under both objectives.” I don't understand why the reward is designed in this way. Why not design it as a similar independent reward and cost like MACPO? Since this paper mainly focuses on RL in continuous time, it may be helpful to highlight the contributions of this paper by separating the reward and cost. ③ Section 4.3 lists the impact of the choice of z on performance changes, but lacks a visualization of how the z value changes over time. I think it would be useful to add such an experiment to show that the changes in z do indeed affect the final decision.

**Reference**

[1] Zhang, Songyuan, et al. "Solving Multi-Agent Safe Optimal Control with Distributed Epigraph Form MARL." arXiv preprint arXiv:2504.15425 (2025).

**Questions:**

Please refer to the “Weakness” section for related questions.

**Details Of Ethics Concerns:**

Regarding ethical review, I have no concerns.

---

> ### Author Response · Authors · 2025-11-24
> **We apperciate the valuable comments from the reviewer, the detailed responses are listed below: Reply 1**
>
> We thank the reviewer for the valuable feedback and constructive suggestions. We provide one-by-one response below.
>
> **W1. Differences between this work and [1].**
>
> While our work is inspired by the epigraph-based formulation, [1] and our paper differs substantially in problem formulation, theoretical development, and algorithmic design.
>
> (1) The paper [1] is fully discrete-time and relies on a discrete Bellman equation. In contrast, our work addresses the continuous-time setting and develops an epigraph formulation compatible with the HJB equation. It is not just a trivial extension, it requires theoretical adaption for the epigraph technique since it cannot be directly applied to standard HJB equations. Our Lemma 3.1 and Theorem 3.3 establish new epigraph-based HJB relationships that do not appear in [1].
>
> (2) The method of two papers is fundamentally different. [1] uses a discrete value-iteration-style update with standard function approximation. While our approach employs a PINN-based critic and other continuous-time technique like value gradient iteration (VGI). These techniques are specifically designed for solving the problem in CT-CMDP.
>
> (3) We also make some adaptations to the epigraph mechanism compare with [1]. [1] separates the optimization into an inner update during training and an outer root-finding procedure during execution. In contrast, we incorporate both optimizations inside training, eliminating the costly online root-finding step and leading to more stable training process.
>
> In summary, while related, our method is not a direct extension of [1]; The continuous-time formulation results in essential differences in both problem structure and solution methodology.
>
> **W2. Methodology Questions.**
>
> **(1) Clarification of Fig. 1.**
>
> We appreciate the reviewer for pointing out this typo. We updated this figure by adding an arrow.
>
> **(2) Transition from Eq. (7) to Eq. (8) and (9).**
>
> Based on Eq. (7), Eq. (8) makes this definition explicit by substituting the
> neural-network approximations $V_\phi^{\text{cons}}(x)$ and $V_\psi^{\text{ret}}(x)$ into
> $V_{\phi}^{\text{cons}}(x) = \sup_{\tau \ge t} c(x(\tau))$, and
> $V_{\psi}^{\text{ret}}(x) = \int_{t}^{\infty} \gamma^{\tau-t}l(x(\tau),\pi(\tau))d\tau,$
> identifying the corresponding  $z^*(x) $.
>
>  Eq. (9) then describes how the resulting $z^*(x)$ is used to construct the network-based approximation of the epigraph-based value. We have added additional explanations in the revised manuscript (highlighted in blue) to improve readability
>
> **(3) Optimization of the three critic losses.**
> Yes, these three losses are optimized jointly. Their weights are chosen so that the losses remain on a similar scale (typically around Residual loss = $0.4$, Target loss = $0.4$, VGI loss = $0.2$), and we select these weights using the grid search. Besides, the impact of different weight configurations is empirically evaluated in our ablation study (section 4.2 Figure 5 and 7). We clarified this joint optimization and include the weight selection strategy in the revised manuscript.

---

> ### Author Response · Authors · 2025-11-24
> **Reply 2**
>
> **W3. Experiments Questions.**
>
> **(1) How MPE and MuJoCo are adapated to continuous-time.**
>
> We agree that it is more important to highlight the differences from their discrete-time counterparts. We have therefore revised Appendix C.1 and C.2 (highlighted in blue) to clearly explain these differences.
>
> Specifically, the original MPE environment updates positions and velocities
> using a fixed step size $\Delta t$ (typically $0.1$), through
> $p_{t+1}=p_t+v_t\Delta t$ and $v_{t+1}=v_t+f\Delta t$. In our continuous-time MPE, we keep the same physical dynamics and integration scheme, but we expose $\Delta t$ as a variable parameter rather than a fixed constant. During training, the algorithm may request any $\Delta t$, and the environment integrates
> $p \leftarrow p + v \Delta t,
> v \leftarrow v + \tfrac{f}{m}\Delta t$
> using the requested step.
>
> In the standard MuJoCo tasks, each environment step corresponds to a fixed number of internal physics frames (e.g., *frame_skip*=5), where a fixed frame interval set as $0.01$. Our continuous-time variant does not execute a fixed number of internal physics frames per environment step. Instead, for any user-specified time interval $\Delta t$, we perform
> $N = \frac{\Delta t}{0.01}$ frames within each step.
> For any prescribed
> $\Delta t$, we execute
> do_simulation(a, N),
> so the effective control interval is entirely determined by the requested $\Delta t$.
>
> **(2) Clarification on the reward design.**
> We want to clarify that we do not introduce a new reward definition, nor do we modify the environment's native metrics. The reward used in our experiments is exactly the one provided by each environment (e.g., distance to the target plus collision penalties in MPE), which is the same structure adopted in MACPO and other constrained-RL baselines.
>
> Our intention of the original narrative was to highlight that, for analysis purposes of the epigraph formulation, we separate the environment reward into (i) the state constraints component (e.g., the collision penalty) and (ii) the task-related cost (e.g., the distance to the target). The sentence “summation of the minus cost and constraints” was therefore misleading, and we have revised the manuscript accordingly in section 4.2 Q1 (highlighted in blue).
>
> Besides, the results that constraint and cost terms are disentangled are also provided in Figure 3 and 5.
>
> **(3) How the z value changes over time.**
> We thank the reviewer for the suggestion. To better illustrate the behavior of the epigraph variable, we visualized the evolution of $z^\ast$ (within one episode) across different stages of training in the MPE--Target task. As shown in Figure 13 (Appendix D.4), early episodes frequently violate the state constraints, causing the constraint branch to remain active and forcing $z^\ast$ to take the clipped upper bound $z_{\max}$. As training progresses and the critic learns to keep
> $V_{\mathrm{cons}}(x_t) \le 0$, the return branch becomes active more
> consistently, and $z^\ast$ decreases smoothly along the rollout. These observations match the expected epigraph definition and confirm that the learned $z^\ast$ meaningfully reflects policy adaptation.

---

> > ### Comment · Reviewer_GrMo · 2025-11-28
> > **Thank you very much for your response. It has helped me gain a clearer understanding of this work. However, as of now, I still have a few points that remain unclear.**
> >
> > **My main concerns are primarily centered on Reply 2.**
> > 1. Regarding Reply 2, specifically the point “**(1) How MPE and MuJoCo are adapted to continuous-time**”: to be honest, your explanation has made this even more confusing to me. I checked the latest submitted version of the paper, particularly the part titled *“Difference from the original discrete-time MPE”* (Line 1048). I do not understand what $p_t$ and $v_t$ represent, nor do I see clearly what the actual difference is. Given that this work emphasizes theoretical analysis, a clearer explanation of the experimental setup might better help assess whether this aspect has been properly validated.
> >
> > 2. Lastly, would it be possible to include **a small additional experiment** demonstrating that standard baseline methods cannot be directly applied to the proposed continuous-time setting? Such a comparison could provide a more intuitive understanding of the distinction between the original and the adapted scenarios.
> >
> > 3. Additionally, I still do not understand the statement: *“the results that constraint and cost terms are disentangled are also provided in Figure 3 and Figure 5.”* Why not plot the cost and reward together in the same figure, using two separate rows or subplots? This could make the claimed disentanglement much more visually intuitive.

---

> ### Author Response · Authors · 2025-11-29
> **Thank you for your reply.**
>
> We sincerely thank the reviewer for carefully reviewing our responses. Your comments greatly help improve the clarity and quality of our paper.
>
> **1. How MPE and MuJoCo are adapted to continuous-time**
>
> We thank the reviewer for pointing this out, and we will revise the text to ensure the explanation is clear.
>
> In the **original discrete-time MPE**, each step updates the state as
> $$
> p \leftarrow p + v\,\Delta t,\qquad
> v \leftarrow v + \frac{F}{m}\,\Delta t,
> $$
> where $p$ is position, $v$ is velocity, $F$ is the action-induced force, and
> $\Delta t$ is a **fixed constant** (typically $\Delta t=0.1$).
>
> To adapt MPE to continuous time, we introduce a new API into the envrionment that changes $\Delta t$ as an **unfixed** argument instead of using the fixed value $\Delta t=0.1$.
>
> For clarity, the update used in the original environment is fix the $\Delta t = 0.1$:
>
> $$
> \texttt{step}(F):\quad
> p \leftarrow p + v \cdot 0.1,\quad
> v \leftarrow v + \frac{F}{m} \cdot 0.1.
> $$
>
> Our continuous-time version introduces:
>
> $$
> \texttt{continuous step}(F, \Delta t):\quad
> p \leftarrow p + v \cdot \Delta t\,(\text{depends on input }\Delta t),\quad
> v \leftarrow v + \frac{F}{m} \cdot \Delta t\,(\text{depends on input }\Delta t).
> $$
> so that the state update depends directly on the argument $\Delta t$ rather than a fixed constant.
>
> MuJoCo internally simulates physics at a fixed frame duration of $0.01$.
> In the standard discrete-time setting, one environment step corresponds to a fixed
> $N = 5 \quad  \text{frames},$ producing an effective step size $\Delta t = 5 \times 0.01 = 0.05$.
>
> Our continuous-time adaptation replaces this fixed number of frames with a variable one.
> (We choose $\Delta t$ to be integer multiples of $0.01$ for numerical stability.)
> The original update is:
>
> $\texttt{step}(u): \quad
> N = 5,\quad \texttt{do simulate}(u, N).$
>
> Our continuous-time version becomes:
>
> $
> \texttt{continuous step}(u,\Delta t): \quad
> N = \Delta t / 0.01,\quad
> \texttt{do simulate}(u, N).
> $
>
> **2.  Additional experiment demonstrating that standard baseline methods cannot be directly applied to the proposed continuous-time setting**
>
> To better illustrate whether discrete-time baselines can perform well in our continuous-time settings, we add up the Figure 15 in Appendix D.6. Specifically, all baselines are adapted to the
> discrete-time setting by removing their residual-loss components.
> Apart from this modification, all implementation
> details follow their original published versions. Across both tasks, EPI consistently achieves lower mean distance to the
> target and smaller variance, demonstrating the performance gain from the modules that designed for the continuous-time settings.
>
> Furthermore, prior work in CTRL has also analyzed why standard discrete-time algorithms degrade when deployed in continuous-time regimes [3, 4]. Their findings align with our empirical observations.
>
> [3] Tallec, Corentin, Léonard Blier, and Yann Ollivier. "Making deep q-learning methods robust to time discretization." International Conference on Machine Learning. PMLR, 2019.
>
> [4] De Asis, Kris, and Richard S. Sutton. "An idiosyncrasy of time-discretization in reinforcement learning." arXiv preprint arXiv:2406.14951 (2024).
>
> **3. Why not plot the cost and reward together in the same figure, using two separate rows or subplots?**
>
> We want to clarify that in our setting (as in standard constrained / safe RL), the reward is combined between cost of the task utility (e.g., the distance to the target in MPE), and penalized
> constraint terms (e.g., the collision penalty for the agents)
>
> Cumulative reward is a direct metric for evaluating the performance since it combined with cost and constraint, like a lot of other papers like MACPO demonstrated. However, to further shows the details of each component like cost or constraint, we want to know if the agents can arrive the target effectively or whether it can avoid collision between each agents or obstacles, we separate the cost and constraint, and visualize them both. This is also a standard and common metric adopted by other safe RL papers ([1] [2]}
>
> [1] Zhang, Songyuan, et al. "Solving Multi-Agent Safe Optimal Control with Distributed Epigraph Form MARL." arXiv preprint arXiv:2504.15425 (2025).
>
> [2] Tayal, Manan, et al. "A physics-informed machine learning framework for safe and optimal control of autonomous systems." arXiv preprint arXiv:2502.11057 (2025).

---

### Official Review · Reviewer_VtHr · 2025-11-11

**Soundness:** 3
**Presentation:** 3
**Contribution:** 3
**Rating:** 6
**Confidence:** 3

**Summary:**

The paper formulates safe CT--MARL as a continuous-time CMDP and introduces an epigraph variable $z$ to smooth discontinuities, deriving an epigraph-HJB and training a PINN-based actor--critic with a revised inner/outer optimization that computes $z^\*$ during rollouts. Results are on safe CT-MPE and CT-MA-MuJoCo with ablations.

**Strengths:**

1. Addresses a real gap: explicit continuous-time safe MARL (state constraints) with a principled epigraph--HJB route and viscosity-solution framing.

2. Clean derivations: Lem.3.1/3.2 and Thm. 3.3 tightly connect DPP to the PDE with the $\ln\gamma$ term.

3. Practical training design:  $z^\*$ computed during training; $z$-independent critics simplify deployment.

4. Broad evaluation \& ablations:  consistent gains on CT-MPE/MA-MuJoCo; ablations clarify each loss term’s effect (target/VGI $>$ residual).

**Weaknesses:**

1. Feasibility of the outer problem: Eq.(8) requires $\max\{V _{\text{cons}}(x),V _{\text{ret}}(x)-z\}\le 0$. If $V _{\text{cons}}(x)>0$, no $z$ is feasible; the paper clips $z^\*$ to $[z _{\min},z _{\max}]$ (Sec.~3.2.2), which does not restore feasibility or preserve the theory.

2. Model-bias vs. safety: residual/advantage (Eqs.~10,17) use learned $f_\xi,l_\phi$; PDE guarantees hold for true dynamics but not for the surrogate, and no robustness link to violation rates is provided.

3. “Continuous-time” evaluation clarity: Alg.~1 samples arbitrary decision times and the policy conditions on $\Delta t$, but there is no systematic $\Delta t$-sweep or irregular-step stress test in the main results.

**Questions:**

1. When $V_{\text{cons}}(x)>0$ along a rollout, Eq.(8) has no feasible $z$. What exact rule do you use for $z^\*$ in this case, and why does the resulting objective still correspond to the epigraph--HJB target used in Lem.3.1/Thm.3.3?

2. How is $\partial_z\tilde V$ computed in Eq.~(10) with $z$-independent critics? Do you use a straight-through $-{\bf 1}$ on the return branch or a smooth approximation to $\max$? Any stability tricks near the switching set?

3. Given Eqs.~(16)--(17), can you bound the gap between model-PDE residuals and true residuals, or provide stress tests (dynamics mismatch) linking model error to violation rates?

---

> ### Author Response · Authors · 2025-11-24
> **Many thanks to your valuable comments, we provided detailed responses below: Reply 1**
>
> We sincerely thank the reviewer for your valuable feedback and careful reading of our paper. We address the reviewer's questions point by point below.
>
> **W1. Feasibility of the outer problem: no $z^*$ feasible if state constraints are violated.**
>
> We appreciate the reviewer’s insightful observation. We would like to clarify that the theory still holds when $V_{\mathrm{cons}}(x) > 0$. (1) In the epigraph reformulation, when the constraint value is positive (i.e., CMDP state constraint is violated), it is expected that
> $\max\{V_{\mathrm{cons}}(x), V_{\mathrm{ret}}(x)-z\} > 0
> \ \text{for all } z .$
> This behavior is inherent to the epigraph construction: once $V_{\mathrm{cons}}(x) > 0$, the state is already infeasible in the original CMDP, and therefore no $z$ should satisfy $V(x,z)\le 0$, which will lead to $z^{\ast}= + \infty$. However, in practice,  we work with a bounded search interval $z\in[z_{\min},z_{\max}]$ as we stated at the beginning of section 3.2.2. Therefore, the rule above is realized by clipping
> $$
> z^* = \mathrm{clip}\big(z^*, z_{\min}, z_{\max}\big)=\mathrm{clip}\big(+\infty, z_{\min}, z_{\max}\big)=z_{\max},
> $$
>
> This is not a flaw of the theory but exactly how the epigraph is defined. When the constraint is violated during training, the resulting epigraph value naturally takes $z_{\max}$, corresponding to the worst-case (maximized) cost. Intuitively, $z_{\max}$ provides the strongest negative learning signal, encouraging the agents to avoid such infeasible states. We further explain this at the beginning of section 3.2.2 (marked blue) to prevent any confusion.
>
> (2) In practice,  $V_{\mathrm{cons}}(x)$ can be positive and inaccurate at the beginning of training. The outer optimization would push $z^\ast$ to $z_{\max}$. After the policy learns to avoid the state constraints, $z^\ast$ naturally converges within $(z_{\min}, z_{\max})$, and clipping is rarely activated as Figure 13 (Appendix D.4) demonstrated.
>
> **W2. Model-Bias will introduce safety problem.**
>
> Our method does rely on the learned dynamics and reward models. To address the reviewer's concern, we add an experiment about the performance under different stochastic noise settings in Figure 11 (see the blue-marked part in Appendix D.2). We perturb the continuous-time dynamics model by injecting Gaussian noise,
> $x_{t+\Delta t}
> = f(x_t, u_t)\Delta t + \varepsilon_t,
> \varepsilon_t \sim \mathcal{N}(0,\sigma^2 I),$
> and evaluate three magnitudes: Low ($\sigma^2=0.1$), Mid ($\sigma^2=0.5$), and High ($\sigma^2=1.0$). As shown in Figure 11 (Appendix D.2), the No-Noise and Low-Noise settings produce nearly identical cost and constraint-violation performance across all tasks. This indicates that the
> PINN-based value approximation can tolerate small local perturbations.
>
> However, Mid and High noise introduce significant deviations in state propagation, and errors can accumulate over time and distort the training signal. Since our method does not include explicit modules to cope with uncertainty (e.g., stochastic HJB or robust optimization), such large disturbances/mismatch degrades the performance. We will continue to explore this direction in future work.
>
> **W3. No $\Delta t$-sweep or irregular-step stress test in the main results.**
> We have added a $\Delta t$-sweep experiment in Appendix D.3 (Fig. 12), where we fixed $\Delta t \in \{0.03, 0.05, 0.10, 0.15\}.$ For each fixed $\Delta t$, we roll out full trajectories using the learned policy and measure the average distance to the target. Across all three tasks, we observe the task accuracy decreases as $\Delta t$ increases. This is because small $\Delta t$ provides high temporal resolution and enables fine-grained control signals. In contrast, larger $\Delta t$ results in coarser control updates. Moreover, both the HJB residual and the VGI update rely on local differential information; as $\Delta t$ grows, the mismatch between the continuous-time formulation and the discrete rollout increases, amplifying approximation errors in the learned value and value gradients.
>
> **Q1. What exact rule do you use for $z^*$ if the state constraints are violated.**
>
> When $V_{\mathrm{cons}}(x)>0$ along a rollout, we set $z^\ast = z_{\max}.$
> A more detailed explanation is provided in our response to Comment **W1**. Please refer to that discussion for completeness.

---

> ### Author Response · Authors · 2025-11-24
> **Reply 2**
>
> **Q2. How is $\partial_z \hat V(x,z)$ computed in Eq. (10) with independent critics? Any smooth techniques?**
>
> Indeed, the term $\partial_z \hat V$ is computed analytically as
>
> $\partial_z \hat V(x,z) =
> \{-1 \text{(constraints branch inactive)}\; 0 \text{(constraints branch active)}\}$
>
> and we do not use any straight-through estimator or smooth surrogate for the
> $\max$ operator. Although the operator is non-smooth, instability does not occur: Usually, if the gradient oscillate frequently, the switching set has to close to $V_{\text{ret}} - z=V_{\text{cons}}$.
> But in practice, this scenario rarely happens. Specifically, switching between branches is gradual rather than abrupt: when the agent is far from violating constraints, the return branch remains active for many iterations; when approaching constraint boundaries, the constraint branch continues to take over and last for some learning rounds. Therefore, once a branch becomes active, its gradient contribution ($0$ or $-1$) remains constant over many updates, making the learning smooth rather than erratic.
>
>
> **Q3. Can you bound the gap between model-PDE residuals and true residuals, or provide stress tests (dynamics mismatch) linking model error to violation rates?**
>
> We provide the stress tests in Figure 11 (Appendix D.2), highlighted in blue for ease reference. The analysis is the same as the response of W2. Please refer it for the completeness.

---

### Meta-Review · Area_Chair_Urcm · 2026-01-05

**Summary:**

This paper studies continuous time safe MARL. To deal with the non-smoothness introduced by the constraint violation, an epigraph based method is proposed. The major contribution is the extension to the continuous time setting through the transformation.

**Reviewer Concerns:**

Reviewer VtHr concerns about some technical details of the paper, including feasibility issue and algorithm implementation.
Reviewer GrMo questions about the connection to a prior work and some questions on experiments.
Reviewer qGRY concerns about experiment details.
Reviewer cjAT questions about the details and motivations.

After reading authors' rebuttal, I believe most of these concerns are largely addressed. The authors also include extensive new results, which I found very effective.

**Reviewer Scores:**

I believe reviewers' concerns are largely addressed.

---

### Decision · Program_Chairs · 2026-01-26

Accept (Poster)